# Bridged filaments of histone-like nucleoid structuring protein pause RNA polymerase and aid termination in bacteria

Matthew V Kotlajich[1], Daniel R Hron[1], Beth A Boudreau[1], Zhiqiang Sun[2], Yuri L Lyubchenko[2], Robert Landick[1,3]*

[1]Department of Biochemistry, University of Wisconsin–Madison, Madison, United States; [2]Department of Pharmaceutical Sciences, University of Nebraska Medical Center, Omaha, United States; [3]Department of Bacteriology, University of Wisconsin–Madison, Madison, United States

**Abstract** Bacterial H-NS forms nucleoprotein filaments that spread on DNA and bridge distant DNA sites. H-NS filaments co-localize with sites of Rho-dependent termination in *Escherichia coli*, but their direct effects on transcriptional pausing and termination are untested. In this study, we report that bridged H-NS filaments strongly increase pausing by *E. coli* RNA polymerase at a subset of pause sites with high potential for backtracking. Bridged but not linear H-NS filaments promoted Rho-dependent termination by increasing pause dwell times and the kinetic window for Rho action. By observing single H-NS filaments and elongating RNA polymerase molecules using atomic force microscopy, we established that bridged filaments surround paused complexes. Our results favor a model in which H-NS-constrained changes in DNA supercoiling driven by transcription promote pausing at backtracking-susceptible sites. Our findings provide a mechanistic rationale for H-NS stimulation of Rho-dependent termination in horizontally transferred genes and during pervasive antisense and noncoding transcription in bacteria.

*For correspondence: landick@biochem.wisc.edu

**Competing interests:** The authors declare that no competing interests exist.

## Introduction

Important features of cellular regulatory programs depend on interactions between the transcriptional machinery and DNA packaged in nucleoprotein complexes in vivo. The impact of nucleoprotein on transcriptional regulation has been elucidated in greatest detail in eukaryotes, where nucleosome structure and dynamics affect efficient initiation complex assembly (*Knezetic and Luse, 1986*; *Lorch et al., 1987*; *Li et al., 2007*), affect transcript elongation by RNA polymerase II (RNAPII) (*Studitsky et al., 1997*; *Kireeva et al., 2005*; *Bintu et al., 2012*; *Kulaeva et al., 2013*), and, conversely, are modulated by factors associated with elongating RNAPII (*Kristjuhan and Svejstrup, 2004*; *Workman, 2006*). However, our understanding of the impact of nucleoprotein on transcription in bacteria is more primitive, principally because the structures of nucleoprotein complexes formed from DNA and nucleoid-associated proteins (NAPs) are more heterogeneous in structure, more dynamic, and less stable than nucleosomes. The histone-like nucleoid structuring protein (H-NS) is the principal NAP in *Escherichia coli*. H-NS binds DNA at high-affinity sites, spreads to form filaments on AT-rich DNA, bridges between filaments on different DNA segments, and inhibits transcription (reviewed in *Ali et al., 2012*; *Dorman, 2007, 2009*; *Navarre, 2010*; *Navarre et al., 2007*).

H-NS is present at ~2 × 10⁴ copies per cell (*Ali Azam et al., 1999*), enough to cover ~14% of a single-copy genome, as currently modeled in bridged filaments (*Arold et al., 2010*). ChIP experiments in *E. coli* and *Salmonella* reveal sequestration in H-NS filaments of ~350 DNA segments 0.5–50 kb in length (~2 kb on average) that change little in different growth or environmental conditions and

**eLife digest** Genes—which are made of DNA—contain the genetic blueprint of an organism. Different genes are switched on (expressed) at various points in an organism's life when they are needed. When a gene is switched on or 'expressed', the DNA is copied using molecules of ribonucleic acid (RNA) that can then be used as templates to make proteins. One way that the expression of genes can be controlled is by the way that DNA is packaged in cells.

In humans and other eukaryotic organisms, DNA is packaged within groups or 'complexes' of proteins called histones. The histones can interact with the cellular machinery that moves along the gene and makes the RNA copies. In doing so, the histones can alter the length of the RNA copies, and if the RNAs are too short, they will not be used to make proteins so the gene is effectively switched off.

We know much less about how DNA packaging in bacteria cells affects gene expression. There are many different DNA packaging complexes in bacteria and they are less stable than the histones in eukaryotes, which makes them harder to study. In the bacteria *Escherichia coli*, the main DNA packaging protein is the histone-like structuring protein (H-NS). This protein binds to the DNA at select sites and forms filaments that can link to each other to form bridges between different stretches of DNA.

In this study, Kotlajich et al. found that the bridged filaments formed by the H-NS protein interfere with the production of RNA copies. It is normal for the enzyme that makes RNA copies from DNA—called RNA polymerase—to make short pauses while it moves along the DNA. However, the bridged filaments made by H-NS cause RNA polymerase to pause for longer periods of time. These delays provide more time for another protein that halts gene copying to bind to the site where the RNA polymerase has paused, leading to RNA molecules that are too short to make proteins.

It has previously been shown that there is less bridging between H-NS filaments in some bacteria when they are grown in warmer temperatures—around 37°C—than when they are grown in cooler temperatures of around 20–30°C. This may allow bacteria that cause diseases in animals to increase the expression of genes that help them outwit the host's defenses and to resist antibiotic treatments when they enter a warm animal body from a colder environment.

correlate with higher AT-content and reduced gene expression (**Oshima et al., 2006**; **Noom et al., 2007**; **Vora et al., 2009**; **Kahramanoglou et al., 2011**; **Peters et al., 2012**; **Myers et al., 2013**). A large fraction of these filaments co-localize in clusters (~2 clusters per chromosome in *E. coli*) that likely depend on H-NS bridging (**Wang et al., 2011**), although the extent and time-scale of bridging rearrangements is unknown.

The 15.5-kDa H-NS monomer consists of an N-terminal oligomerization domain with two oligomerization sites (head and tail; *Figure 1—figure supplement 1*) separated by a 45-aa α-helical linker; a 46-aa C-terminal DNA-minor-groove-binding domain connects to the oligomerization domain through a 10-aa flexible linker (**Shindo et al., 1999**; **Arold et al., 2010**; **Cordeiro et al., 2011**; **Gordon et al., 2011**). H-NS lacking DNA-binding domains forms helical proteinaceous filaments with head–head and tail–tail interfaces (**Arold et al., 2010**). H-NS binds DNA at discrete high-affinity sites with site-specific regulatory function (**Bouffartigues et al., 2007**; **Lang et al., 2007**). Depending on surrounding sequence, available H-NS, and other factors (temperature, solute composition, other NAPs), filaments form by spreading (**Amit et al., 2003**; **Bouffartigues et al., 2007**; **Cordeiro et al., 2011**). Current models suggest that the DNA-binding domains of one tail–tail module bind adjacently to a single DNA segment in linear filaments, which form at <5 mM Mg$^{2+}$, or contact separate DNAs or DNA segments in bridged filaments favored at >5 mM Mg$^{2+}$ (**Liu et al., 2010**).

H-NS filaments (and associated NAPs) silence transcription of horizontally transferred DNA (**Navarre et al., 2007**) and suppress pervasive noncoding and antisense transcription (**Peters et al., 2012**; **Singh et al., 2014**; **Wade and Grainger, 2014**) by controlling RNAP initiation at promoters (**Dame et al., 2002**; **Fang and Rimsky, 2008**; **Dorman, 2009**; **Singh et al., 2014**), by inhibiting transcript elongation by RNAP (**Dole et al., 2002, 2004**; **Saxena and Gowrishankar, 2011**; **Peters et al., 2012**), or both. For both silencing and suppression of antisense and noncoding transcription, in vivo

experiments indicate that H-NS filaments slow or block elongating RNAP, although direct biochemical tests of elongating RNAP–H-NS interactions or insights into the underlying mechanisms have not been reported. In vivo assays also establish that the H-NS block to transcription is greater in enterobacteria growing at 20–30°C outside hosts than at 37°C typical for symbiotic or pathogenic growth, and implicate H-NS in switching gene expression upon host invasion (*Goransson et al., 1990*; *Trachman and Yasmin, 2004*; *Ono et al., 2005*; *Yang et al., 2005*).

Rho-dependent termination also plays a key role in suppressing both horizontally transferred genes and pervasive noncoding transcription (*Saxena and Gowrishankar, 2011*; *Nicolas et al., 2012*; *Peters et al., 2012*; *Singh et al., 2014*). In both cases, a strong association between sites of Rho-dependent termination and sites of H-NS filament formation implicates H-NS in delaying transcript elongation to aid Rho in dissociating elongating RNAP from DNA and preventing synthesis of RNAs potentially deleterious to the cell (*Ali et al., 2012*; *Peters et al., 2012*; *Singh et al., 2014*).

To investigate whether H-NS filaments pose direct barriers to transcript elongation and to characterize underlying mechanisms, we focused on an antisense transcription unit in the well-characterized *bgl* operon of *E. coli* K-12. *bglGFB* encodes cryptic genes for ß-glucoside catabolism and is ordinarily silenced by H-NS filaments that emanate from high-affinity sites flanking the promoter (upstream regulatory element, URE, and downstream regulatory element, DRE, respectively; *Figure 1A*). H-NS filaments nucleating on the DRE block RNAP in both the sense and antisense directions (*Dole et al., 2004*; *Peters et al., 2012*). Using in vitro transcription and direct visualization of H-NS filaments and elongating RNAP by atomic force microscopy (AFM), we found that H-NS filaments directly inhibit elongating RNAP and promote Rho-dependent termination, but surprisingly only when H-NS forms bridging interactions.

## Results

### High H-NS–DNA ratio switches bridged nucleoprotein filaments to linear filaments

To investigate how H-NS filaments affect transcript elongation by RNAP, we engineered a transcription template that could form $\lambda P_R$ promoter-initiated, halted A26 elongation complexes (ECs) on a 26 nucleotide C-less cassette placed upstream from the *bgl* DRE and URE in the antisense direction (*Figure 1A*). On this template, $\lambda P_R$ drives synthesis of the *bgl* antisense transcript from a position ~1500 bp closer to the DRE than the *bglF* antisense promoter. Initial generation of halted ECs allowed us to uncouple transcription initiation from subsequent H-NS filament formation and transcript elongation, a key experimental feature impossible for in vivo studies.

To understand filament formation on our *bgl* template, we examined H-NS–DNA interactions by native PAGE at 8 mM $Mg^{2+}$, a condition previously found to favor bridging interactions (*Figure 1A*). At low DNA concentration (10 pM), we observed half-maximal retardation of radiolabeled DNA electrophoresis at ~2 nM H-NS, which we infer corresponds to either the $K_d$ of H-NS nucleation on the high-affinity URE and DRE sites or the $K_d$ for filament extension. This affinity of H-NS is tighter than previously reported (*Azam and Ishihama, 1999*; *Dole et al., 2004*), perhaps because prior measurements required >2 nM H-NS to form stable filaments at the DNA concentrations used or because, to mimic in vivo conditions, we substituted glutamate for chloride often used previously. As the H-NS concentration was increased, we observed two distinct complexes at both 10 pM and 10 nM DNA. A slower migrating band was observed at 66 H-NS/kb (for 10 nM DNA). At higher concentrations of H-NS (80–200 H-NS/kb), the complexes migrated more rapidly, producing a visible gel downshift. Both the faster and slower migrating species were detected at both low and high DNA concentrations; however, it took higher H-NS:DNA ratios to achieve the comparable protein–DNA complex shifts at 10 pM DNA (333 H-NS/kb for the slower and $3.3 \times 10^4$ H-NS/kb for the faster migrating species). With 10 nM DNA, we found that the shift in gel mobility occurred at both low (2 mM) and high (8 mM) $Mg^{2+}$ when H-NS concentration was increased from 66 H-NS/kb to 200 H-NS/kb either using radiolabeled DNA alone or using DNA containing halted A26 ECs with radiolabeled RNA (*Figure 1B*).

We hypothesized that the differences in the filament gel mobility resulted from a switch from bridged to linear filaments as the concentration of H-NS was increased. To test this idea, we examined the filaments using AFM. We observed four different H-NS-induced topologies, which we classified linear, interwound, circular, and hairpin based on their appearances (*Figure 1C* and *Figure 1—figure supplement 2D*). On linear filaments, H-NS spread over the entire DNA molecule but interactions

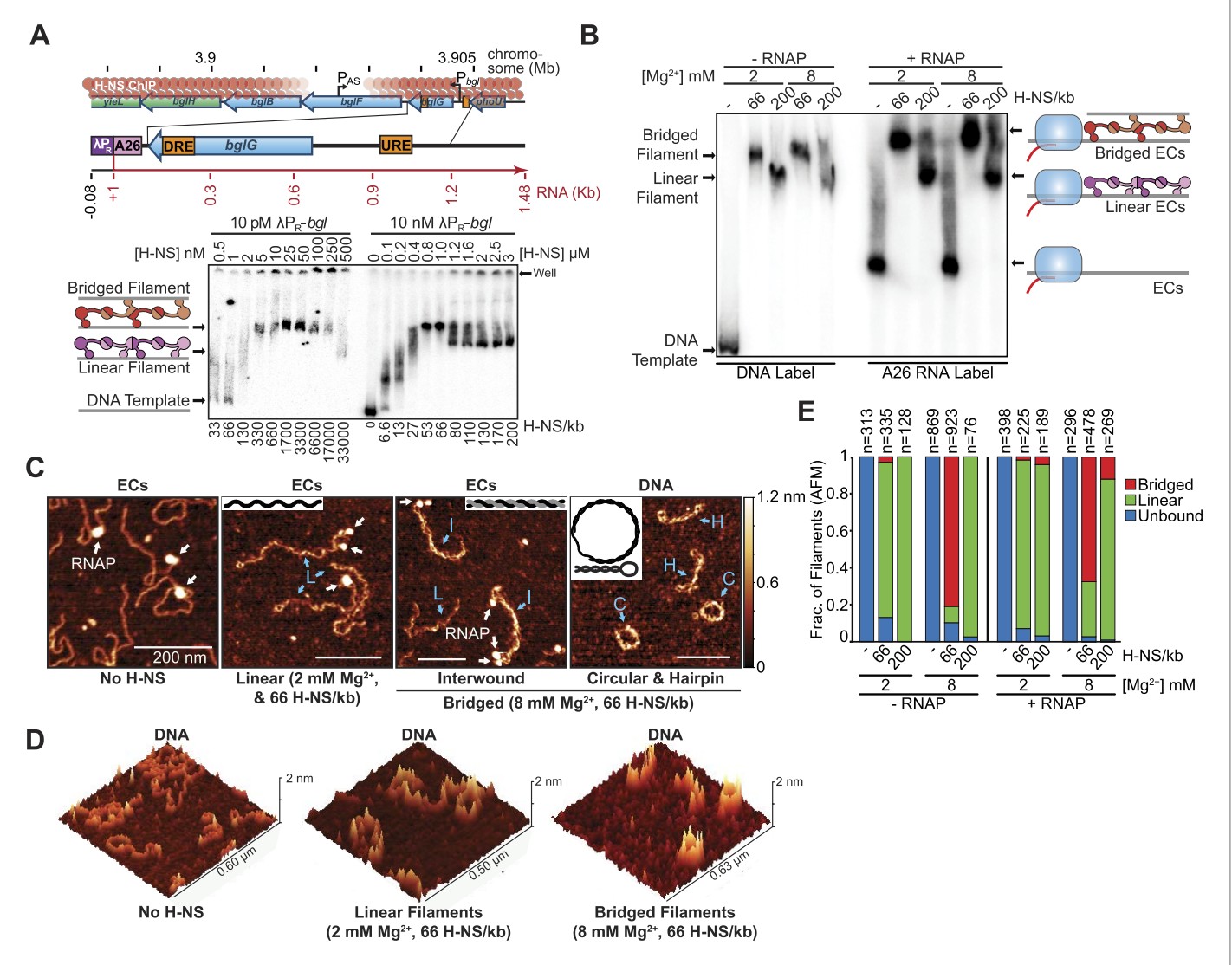

**Figure 1**. H-NS formed two different filaments depending on concentration. (**A**) The H-NS-silenced *E. coli* bgl operon, encoding genes for ß-glucoside catabolism, contains an antisense promoter within *bglF* (P_AS) (***Peters et al., 2012***). The 1.56-kb linear λP_R-*bgl* DNA template contains the λP_R promoter followed by a 26-nucleotide C-less cassette (to allow formation of halted A26 ECs) and two high affinity H-NS binding sites (DRE and URE) (***Dole et al., 2004***). Native PAGE of H-NS filaments on 10 pM or 10 nM λP_R-*bgl* template in 8 mM Mg²⁺. Graphics depicting bridged and linear H-NS filaments are shown left of the gel and related to H-NS molecular structures in ***Figure 1—figure supplement 1***. (**B**) Native PAGE of H-NS filaments formed on free DNA or halted A26 complexes (at 10 nM) at 2 or 8 mM Mg²⁺ and 66 H-NS/kb or 200 H-NS/kb (1 or 3 µM H-NS respectively). ³²P-labeled DNA (10 nM) was used for lanes denoted –RNAP; unlabeled DNA and ³²P-labeled RNA formed by incorporation of [a-³²P]GTP were used for lanes denoted +RNAP. (**C**) Representative AFM images of H-NS filaments on DNA or ECs matching the EMSA assays shown in (**B**). DNA or ECs with either 66 H-NS/kb or 200 H-NS/kb were diluted from 10 nM to 2 nM, immediately absorbed on APS-mica, and imaged in air. RNAP bound to DNA is indicated by white arrows. Cyan arrows indicate linear H-NS complexes (L), interwound H-NS complexes (I), circular H-NS complexes (C), or hairpin H-NS complexes (H). Graphics depicting the observed DNA topologies are shown in insets, where gray or black lines are each equivalent to one dsDNA molecule. AFM images from which these panels were cropped and additional examples are shown in ***Figure 1—figure supplement 2***. (**D**) Pseudo-3D images of complexes similar to those in panel **C**, but lacking ECs to avoid scaling distortion. (**E**) Complexes were binned based on their DNA topology defined in (**C**). Interwound, circular, and hairpin H-NS complexes were grouped together as various forms of bridged complexes. H-NS complexes formed on template DNA are denoted –RNAP, and +RNAP denotes complexes formed on ECs. Only complexes with RNAP bound were counted in the +RNAP samples.

The following figure supplements are available for figure 1:

**Figure supplement 1**. Model of H-NS filaments.

*Figure 1. Continued on next page*

*Figure 1. Continued*

**Figure supplement 2**. Interwound filaments formed preferentially in samples of ECs at 8 mM Mg²⁺ and 66 H-NS /kb.

**Figure supplement 3**. Temperature affected H-NS bridging interactions.

between DNA segments or with another DNA molecule were not visible. Interwound complexes contained two DNAs, which were sometimes discernible as distinct filaments but generally exhibited increased height above the mica surface compared to linear filaments (*Figure 1D*). Occasionally, linear and bridged filaments were observable in the same AFM field (*Figure 1C*, third panel). Circular and hairpin complexes appeared to have similar bridging to the interwound filaments, but involved only one DNA. We categorized the interwound, hairpin, and circular topologies as different forms of bridged filaments, whereas the remaining bound species were characterized as linear filaments. These different filament topologies were generally consistent with previously reported AFM images of H-NS–DNA complexes (*Dame et al., 2000*; *Maurer et al., 2009*; *Liu et al., 2010*), except that RNAP was apparent in images prepared from samples containing A26 ECs (e.g., *Figure 1C*, panel 1,2,3; *Figure 1—figure supplement 2A*).

To quantify the types of filaments formed in different conditions, we binned images based on their shape, width, contour length, and height ('Materials and methods'). In our transcription conditions (10 nM DNA diluted to 0.5–1 nM at 4°C for AFM imaging), bridged filaments predominated at 8 mM Mg²⁺ and 66 H-NS/kb (~80% of filaments without RNAP and ~70% of filaments with RNAP were interwound, hairpin, or circular; *Figure 1D*), whereas linear filaments predominated at lower Mg²⁺ (2 mM) and higher H-NS:DNA ratio (200 H-NS/kb) at both 2 and 8 mM Mg²⁺. H-NS filaments also exhibited decreased contour lengths (were compacted) and increased persistence lengths (were stiffer), especially in the bridging configuration (*Figure 1—figure supplement 2B,C*). The AFM images were mostly consistent with our hypothesis that the slower migrating band at 8 mM Mg²⁺ in the EMSA assays contained bridged filaments (either with or without RNAP), although the concentration requirements of AFM imaging precluded direct observations at 10 nM DNA. Additionally, the presence of RNAP in halted A26 ECs caused a shift in the conformation of bridged filaments from a near equal distribution of interwound, hairpin, and circular forms without ECs to predominantly interwound when ECs were present (*Figure 1—figure supplement 2E*).

Consistent with predictions that H-NS bridged interactions and transcriptional silencing may be disrupted at higher temperatures (*Arold et al., 2010*), we found that the more slowly migrating band in EMSA assays, which we attributed to bridged filaments, disappeared when gels were run at 37°C (*Figure 1—figure supplement 3*).

We conclude that H-NS forms bridged complexes principally at high divalent cation concentrations when H-NS is present at 50–66 H-NS/kb and forms linear filaments when H-NS is bound at 2 mM Mg²⁺ or at high concentrations of H-NS (see 'Discussion'). The presence of both slower and faster migrating complexes in EMSA assays of samples formed at 2 mM Mg²⁺, where AFM predicts mostly linear filaments, is an inconsistency in our results but could reflect formation of bridged filaments during electrophoresis in the altered environment of the gel. In general, the detection of H-NS–DNA nucleoprotein filaments by EMSA in buffers lacking Mg²⁺ (see 'Materials and methods') may reflect the caging effects of PA gels that could stabilize both the bridged and linear filaments.

## Bridged but not linear H-NS filaments inhibit transcript elongation in vitro

We next assessed how the linear or bridged filaments affected transcript elongation during single-round in vitro transcription by adding NTPs (30 μM each) to halted A26 ECs after H-NS filament formation (*Figure 2A*). Strikingly, transcript elongation at 8 mM Mg²⁺ was dramatically slower on filaments formed at 66 H-NS/kb (conditions favoring bridged filaments) than on DNA alone, but returned to nearly the rate observed on DNA alone on filaments formed under conditions that favor linear filaments (200 H-NS/kb). We converted the gel images to plots of transcript length by densitometry and comparison to size standards, which allowed calculation of mean transcript lengths for each time point (*Figure 2B–D*). To assign precise pause positions, we also compared pause bands to 3'-deoxyNTP-generated ladders using shorter templates and high-resolution gels (*Figure 2—figure supplement 1*;

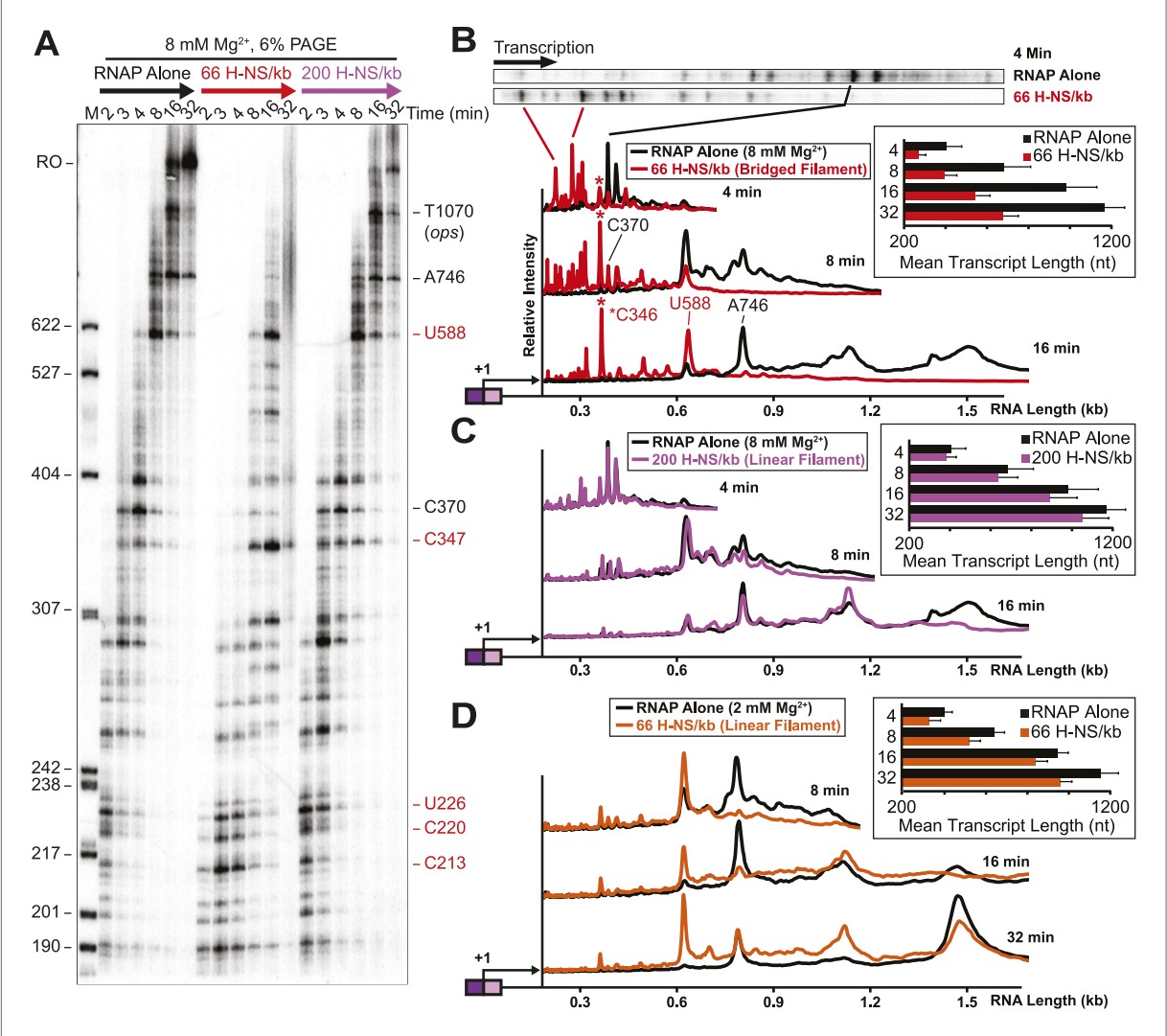

**Figure 2**. H-NS dramatically decreased transcript elongation in vitro. (**A**) In vitro transcription in the presence of 66 H-NS/kb or 200 H-NS/kb filaments at 20°C, 8 mM Mg$^{2+}$, and 30 μM each NTP. ECs (10 nM) were formed at the end of the C-less cassette on the λP$_R$-*bgl* template (A26 ECs) and then incubated with H-NS. Samples were removed at 2, 3, 4, 8, 16, and 32 min after addition of NTPs and separated by 6% PAGE. M, 5′ end-labeled, *Msp*I-digested pBR322 marker. RO, run-off RNA. Pauses mapped to single-nt resolution in *Figure 2—figure supplement 1* and *Table 1* are indicated on the right side of the gel in *red* for H-NS-stimulated pauses and *black* for H-NS independent pauses. (**B**, **C**) Densitometry profiles of transcripts produced at 8 mM Mg$^{2+}$ and 20°C from the λP$_R$-*bgl* template in 66 H-NS/kb or 200 H-NS/kb filaments (**B** and **C**, respectively) or without H-NS (see 'Materials and methods'). In (**B**), the 4-min time point from the gel shown in (**A**) is displayed horizontally to allow alignment with the densitometry profile (larger transcripts are to the right). Key pauses are marked in the profiles. Insets, mean transcript lengths and standard deviations were calculated from at least four independent experiments. (**D**) Densitometry profiles of transcripts produced at 2 mM Mg$^{2+}$ and 20°C from the λP$_R$-*bgl* template in 66 H-NS monomer/kb compared to without H-NS. Inset, mean transcript lengths and standard deviations were calculated from at least four independent experiments.

The following figure supplements are available for figure 2:

**Figure supplement 1**. Mapping of 3′ ends of pauses on λP$_R$-*bgl* template.

**Figure supplement 2**. Linear H-NS filaments had minimal effects on elongation.

**Figure supplement 3**. H-NS effects on transcript elongation also occurred on a different template.

**Table 1.** Pause sites and their responses to H-NS and transcription factors

| Pause position | Pausing | | | | | Termination | | Sequence | |
|---|---|---|---|---|---|---|---|---|---|
| | H-NS | RfaH | GreB | GreA | Rho | Rho + H-NS | Rho + NusG | Pause ↓ | |
| C134 | ↑ | ? | ? | ? | ? | ? | ? | CGCUGAUAACUCAAG**C** | UUUCUUCCUG |
| G162 | ↑ | ↓ | ↓ | ? | – | – | – | AAUUAAGGCUGAACU**G** | AAAUUUUAUU |
| U169 | ↑ | – | ↓ | ↓ | – | – | ↑ | GCUGAACUGAAAUUU**U** | AUUAAUUGCA |
| C213 | ↑ | – | ↓ | ↓ | – | – | – | GCGUGACACCUGCAA**C** | AUCCUCCAUA |
| C220 | ↑ | ↓ | – | ↓ | – | – | – | ACCUGCAACAUCCUC**C** | AUAUUUCCGC |
| U226 | ↑ | ↓ | – | ↓ | – | – | – | AACAUCCUCCAUAUU**U** | CCGCUCAUUU |
| C346 | ↑↑↑ | – | ↓ | – | – | ↑ | – | UAGCUGGAACUCUUU**C** | GGGUAAAGCC |
| C370 | – | ↓ | – | ↓ | ↑ | – | – | CCGCUGGAUAUCCCA**C** | AGCAACGGGU |
| C393, G394 | ↑ | ↓ | ↓ | ↓ | – | ↑ | ↑ | GGUUGGGCAGCAACA**C** | GUUUUGCUGA |
| U588, U589 | ↑↑↑ | – | ↓ | ↓ | ↑ | – | – | UCAAGGCAUACUCUU**U** | UUCUAUUCCA |
| A593 | – | – | – | – | – | – | – | GCAUACUCUUUUUCU**A** | UUCCACUUGA |
| G624 | ↑ | ↓ | ↓ | ↓ | – | – | – | UUCUUUCGCCAGCGC**G** | UUUUUGAAAG |
| G643 | ↑ | – | ↓ | ↓ | – | – | – | UUGAAAGCCAAUUCC**G** | CGCCCCAUGA |
| A746, U747 | – | – | ↓ | – | ↑ | – | ↑ | GCAAGGACCUUUUUU**A** | UAAACAAAAA |
| G926 | ↑ | – | – | – | – | ? | ? | AAUAUGACCAUGCUC**G** | CAGUUAUUAA |
| U996 | ↑ | – | ↓ | ↓ | – | ? | ? | CCAAUAAUUAAGUUA**U** | UGGGAUUUGU |
| U1011 | ↑ | ↓ | ↓ | ↓ | – | ? | ? | UUGGGAUUUGUCUGG**U** | GAAUUAUUUG |
| U1022, U1024 | ↑ | ↓ | ↓ | ↓ | – | ? | ? | GUCUGGUGAAUUAUU**U** | GUCGCUAUCU |
| U1079 (*ops*) | – | – | ↓ | ↓ | ↑ | ? | ? | CUAGUGGCGGUAGCG**U** | GCUUUUUUCA |

Pause positions are given as 3′ RNA nucleotide identity and distance from the transcription start site as mapped by high-resolution PAGE (*Figure 2—figure supplement 1*). ↑, increased pause or termination. ↓, decreased pause or termination. In the sequences shown, pause 3′ ends are bold (under arrow) and the position corresponding to the incoming NTP is underlined.

*Table 1*). These reaction profiles revealed that under bridging conditions H-NS dramatically slowed escape from some pause sites, whereas other pauses remained largely unaffected. Some H-NS pauses were long-lived, with dwell times of more than 12 min (e.g., starred C347 pause; *Figure 2B*).

In contrast, ECs elongating through linear H-NS filaments were only modestly slowed relative to transcription of DNA alone (filaments formed at 200 H-NS/kb in 2 or 8 mM Mg$^{2+}$; *Figure 2C,D*, *Figure 2—figure supplement 2*). Filaments formed at 66 H-NS/kb monomer in 2 mM Mg$^{2+}$ did impede elongation, but to a lesser extent than did filaments formed in bridging conditions (*Figure 2D*, *Figure 2—Figure supplement 2*). Although it is possible that pure linear filaments inhibited transcript elongation to some extent, their effects were clearly less than those of bridged filaments. The inhibitory effect seen at 2 mM Mg$^{2+}$ and 66 H-NS/kb monomer could reflect low or transient levels of bridging in these conditions that were not captured by AFM imaging. We conclude that bridged H-NS filaments dramatically slow RNAP by increasing pausing at a subset of sites and that linear filaments have lesser or no effects on elongating RNAP.

Importantly, the large effects of bridged vs linear H-NS filaments were also observed at physiological NTP concentrations (*Figure 3*, *Figure 3—figure supplement 1*) and at higher temperatures (*Figure 4*, *Figure 4—figure supplement 1*). The effects of bridged filaments persisted to 28°C, but the effects of H-NS largely disappeared at temperatures over 30°C. Thus, H-NS alone inhibits transcript elongation in conditions found during free-living but not inter-host growth of enteric bacteria, consistent with observations that H-NS helps mediate the bacterial temperature response upon infection (*Trachman and Yasmin, 2004*; *Ono et al., 2005*). We also verified that the preferential effects of bridged vs linear filaments on ECs also occurred on a different template on which RNAP initiated transcription at the start site of the *bglF* antisense promoter (*Figure 2—figure supplement 3*). We conclude that H-NS filaments can drastically slow transcript elongation by ECs under physiological conditions that would prevail when *E. coli* grows at ambient temperatures outside a mammalian host.

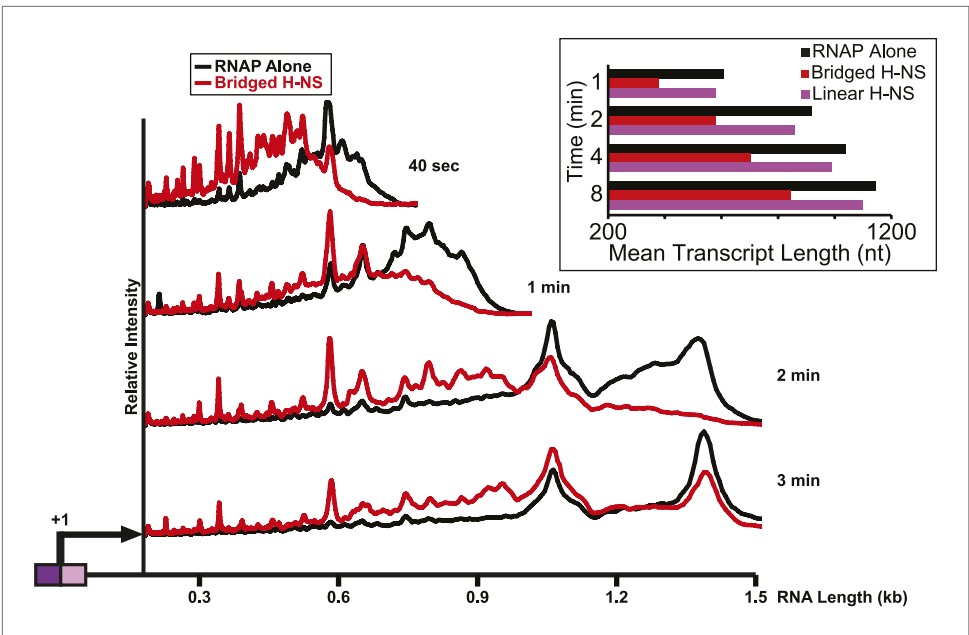

**Figure 3**. H-NS inhibited transcript elongation at physiological NTP concentrations (1 mM each NTP). Densitometry profiles of transcripts produced at 20°C, 12 mM $Mg^{2+}$ and 1 mM each NTP from the $\lambda P_R$-*bgl* template in bridged H-NS filaments (66 H-NS/kb) or in linear H-NS filaments (200 H-NS/kb). Samples were removed at 0.66, 1, 2, 3, 4, 8, and 16 min after addition of NTPs and separated by denaturing PAGE. Inset, mean transcript lengths at various times were averaged from two independent experiments.
The following figure supplement is available for figure 3:

**Figure supplement 1**. Electrophoretic gel image showing H-NS inhibited transcription elongation at physiological NTP concentrations (1 mM each NTP).

## H-NS can bridge around ECs

The preferential inhibition of transcript elongation by bridged vs linear H-NS might reflect either tighter binding of H-NS in the bridging mode or a topological effect of sequestering ECs in a DNA segment flanked by bridged filaments on both sides (see 'Discussion'). To determine whether bridged H-NS filaments readily re-formed upstream of ECs on recently transcribed DNA segments, we used AFM to examine the configuration of ECs and filaments after ECs had transcribed about half the DNA template (8 and 16 min at 30 μM NTPs; *Figure 5A*). H-NS was clearly able to bridge around ECs located in the middle of the DNA template (~80% of observed complexes, n = 93; *Figure 5A* complexes I–X, *Figure 5B*). A minority of complexes (~20%) did not exhibit H-NS bridging on one side of ECs, either because the shorter bridged segments unravel during deposition on AFM slides or because bridged filaments failed to form on newly exposed DNA upstream from elongating ECs (*Figure 5A*, complexes XI and XII).

These observations suggest bridged H-NS may constrain ECs within a topologically closed domain that could promote pausing (see 'Discussion') but do not rule out the possibility that bridged H-NS binds DNA tighter than in linear H-NS filaments to create a stronger roadblock to transcript elongation.

### Bridged H-NS filaments promote pausing at recognizable pause sequences

To investigate the mechanistic basis of H-NS stimulation of transcription pausing, we examined the precise 3' ends of paused transcripts stimulated by H-NS (*Table 1*, *Figure 2—figure supplement 1*). In general, the sites of pausing corresponded to the recently mapped consensus pause sequence for *E. coli* RNAP (*Larson et al., 2014*), with pausing occurring just after pyrimidine addition and just prior to purine addition and with Gs at positions −10 or −11. H-NS increased pausing strongly at a subset of sites (e.g., C346, U588, and some others; *Table 1*) but had less effect at other pause sites recognized

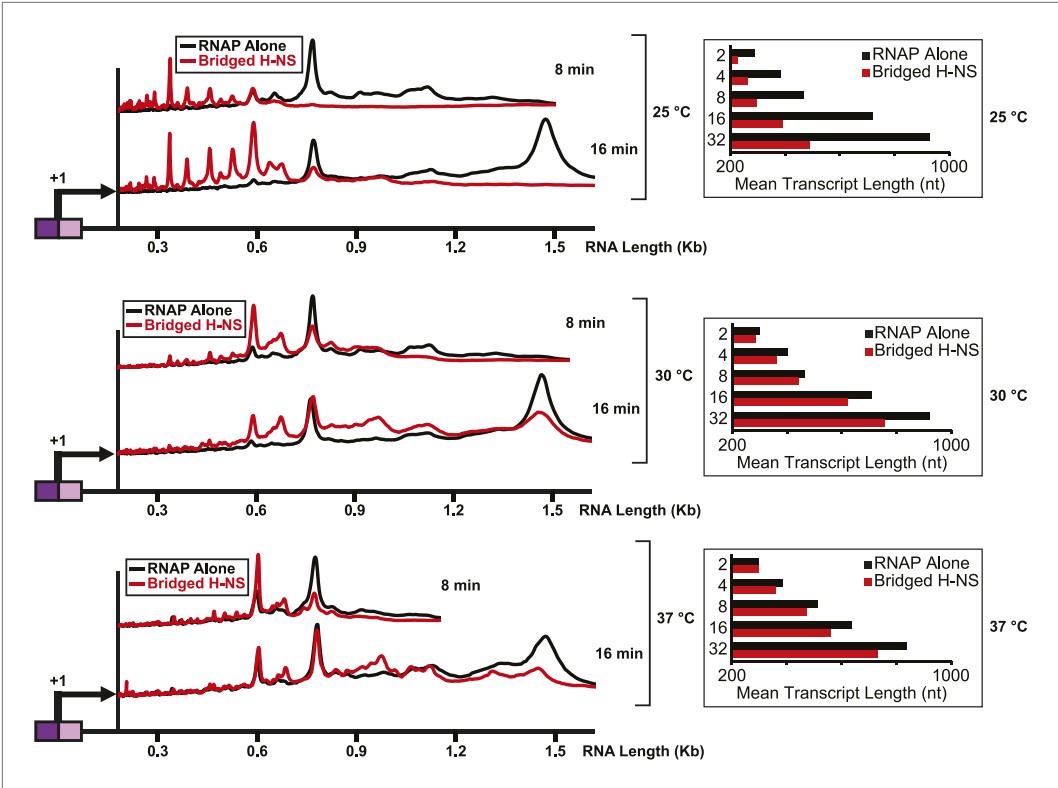

**Figure 4**. H-NS effects on transcript elongation were reduced at ≥30°C. Densitometry profiles of transcripts produced at 25°C, 30°C, or 37°C, 8 mM Mg²⁺ and 30 μM each NTP from the λP$_R$-*bgl* template in the presence of 66 H-NS/kb (bridged filaments). Samples were removed at 2, 4, 8, 16, and 32 min after addition of NTPs and separated by denaturing PAGE. Insets, mean transcript lengths at various time points plotted were averaged from two independent experiments.
The following figure supplement is available for figure 4:

**Figure supplement 1**. Electrophoretic gel image showing reduced H-NS effects on transcription elongation at ≥30°C.

in the absence of H-NS (e.g., at C370 and A746, and at U1079, the *ops* site present on the template). Interestingly, both H-NS-sensitive pauses occurred with Us at positions −2 and −3, whereas 2 of 3 H-NS resistant pauses lacked this feature (C370 and A593). The H-NS-resistant A746 pause containing a U-tract at these positions and the A593 pause were atypical, as they occurred with a 3′ A. 3′ A is known to be highly sensitive to a 1-nt backtrack conformation, which can be readily cleaved by intrinsic hydrolysis and re-extended (*Sosunova et al., 2013*). The 3′-proximal U-tracts observed in the H-NS-stimulated pause sequences should make paused ECs especially sensitive to >1-nt backtracking (*Komissarova and Kashlev, 1997*; *Nudler et al., 1997*), suggesting that pauses sensitive to multiple-nt backtracking may be most strongly affected by H-NS.

## Bridged H-NS filaments promote backtrack pausing by RNAP

To investigate further the nature of H-NS-sensitive pauses, we next examined the effects of RfaH and GreB (*Figure 6*). RfaH and its paralog NusG are thought to suppress entry into pauses by favoring forward translocation and can additionally inhibit hairpin-stabilized pauses by inhibiting RNAP clamp opening through an interaction of their NTDs with the clamp (*Herbert et al., 2010*; *Hein et al., 2014*) (*Figure 6A*). Although NusG can inhibit backtracking to some extent, GreB is a far more effective anti-backtracking factor because it rescues ≥2-nt backtracked ECs by promoting intrinsic transcript cleavage (*Borukhov et al., 1993*; *Laptenko et al., 2003*). However, GreB and its paralog GreA (which promotes cleavage of 1-nt backtracked RNAs) have little effect on the frequency of pausing overall or on hairpin-stabilized pausing (*Feng et al., 1994*; *Artsimovitch and Landick, 2000*).

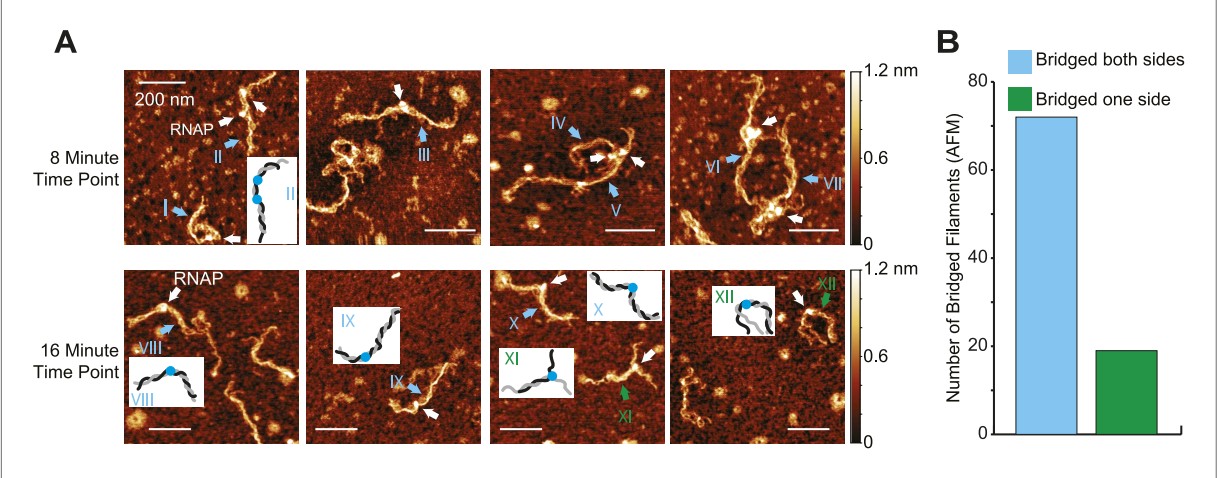

**Figure 5**. Bridged H-NS filaments reformed upstream of ECs during transcription. (**A**) Representative AFM images of ECs elongating through bridged filaments (66 H-NS/kb; 8 mM Mg²⁺; 20°C) sampled at either 8 or 16 min after addition of NTPs (30 μM each). ECs (10 nM) and H-NS filaments were absorbed onto APS-mica and imaged in air. ECs are indicated by white arrows. Roman numerals and arrows depict two classes of filaments formed during transcription (cyan, bridged on both sides of the EC; green, unbridged on one side of the EC). Depictions of ECs and filaments are shown in insets for a subset of panels (black, gray different DNA duplexes; blue, RNAP). (**B**) Quantification of H-NS filament disposition during EC elongation from AFM images like those shown in (**A**). Cyan bar, bridged H-NS filaments both upstream and downstream of ECs. Green bar, bridged H-NS filaments on only one side of ECs.

We tested the effects of RfaH-NTD (which is sufficient for full pause–suppression activity of RfaH), NusG, GreB, and GreA. Transcription through bridged H-NS filaments in the presence of 300 nM RfaH-NTD and 1 mM NTPs resulted in partial suppression of H-NS-stimulated pausing (*Figure 6B,C*). RfaH-NTD ameliorated the effect of H-NS on the C393, G624, A746, G926, U996, U1011, and U1022 pauses but not on the strong H-NS-stimulated C346 and U588 pauses. The effects of RfaH-NTD on individual pauses were more evident than the effect on mean transcript length because RfaH-NTD also caused a strong, H-NS independent pause at the U1079 *ops* site (*Artsimovitch and Landick, 2002*) that dominated the mean transcript calculations. NusG at 100 nM had similar partial effects on overcoming the H-NS-stimulated pauses (*Figure 6—figure supplement 1A,B,C*).

In contrast to these partial effects on H-NS-stimulated pausing, GreA at 500 nM or GreB at 50 nM dramatically reduced the effects of H-NS on elongation (*Figure 6B,D*, *Figure 6—figure supplement 2A,B,D*). All but one pause at least moderately H-NS-stimulated (G926) and both pauses strongly stimulated by H-NS (C346 and U588) were reduced by GreB, whereas the two pauses not affected by GreB (C370 and G926) were suppressed by RfaH NTD (*Table 1*). One pause (C346) was affected by GreB less so by GreA, suggesting was backtracked by multiple nt.

To investigate the difference between pauses strongly affected by H-NS and GreB vs pauses affected by RfaH-NTD but less by H-NS and GreB, we chose the C346 and C370 pauses and engineered scaffolds from the sequences that allowed examination of RNAP behavior with single-nucleotide resolution (*Figure 6—figure supplement 3A,B*). Although scaffolds lack sufficient lengths of duplex DNA to form H-NS filaments, they proved highly informative about the nature of these pauses. The C346 was not detectable at 37°C or 20°C during active elongation (on the fly) but became a strong pause when RNAP was delayed at the C346 position by transiently withholding GTP (delay; *Figure 6—figure supplement 3C,E*). In contrast, the C370 pause was readily detectable on the fly and increased mostly in the fraction paused rather dwell time when delayed at the C370 position (*Figure 6—figure supplement 3D,F*). To understand whether pausing at C346 or C370 involved backtracking, we tested intrinsic cleavage of the nascent RNA in ECs halted at the pause sites. Intrinsic cleavage occurs at internal positions of backtracked transcripts that occupy the RNAP active site and can map the extent of backtracking. The C346 but not the C370 RNA was highly sensitive to intrinsic cleavage in halted ECs; the positions of cleavage suggested that halted C346 ECs spontaneously backtracked by 4–6 nt (*Figure 6—figure supplement 3G*). We confirmed this result by testing susceptibility to GreB cleavage. Again the C346 halted EC was more sensitive than the C370 halted EC

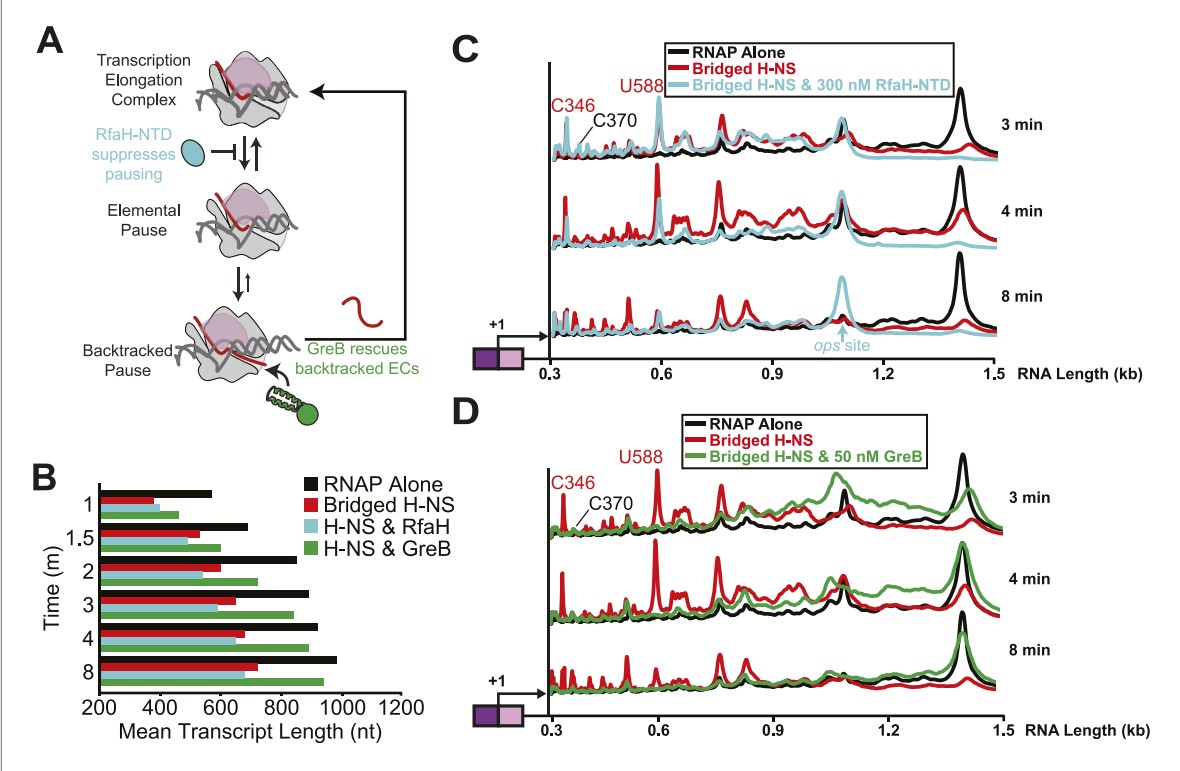

**Figure 6**. Bridged H-NS filaments induced RNAP backtracking, which was rescued by GreB. (**A**) Steps in pausing affected by the NusG-like N-terminal domain (NGN) of RfaH (RfaH-NTD) or Gre factors (e.g., GreB). Binding of the NGN RfaH-NTD (cyan) to the clamp domain (pink) of RNAP inhibits clamp motion and suppresses entry into pause states (*Sevostyanova et al., 2011*). The duration of pausing once paused ECs form can be increased by backtracking of DNA and RNA through RNAP, during which the 3′ RNA enters the RNAP secondary channel. GreB promotes endonucleolytic cleavage of the backtracked RNA in the RNAP active site to convert an offline paused EC back to an active EC (*Laptenko et al., 2003*). (**B**) Mean transcript lengths were averaged from two independent experiments. (**C**, **D**) Densitometry profiles of transcripts produced at 20°C, 12 mM $Mg^{2+}$, and 1 mM each NTP from the $\lambda P_R$-*bgl* template in bridged filaments (66 H-NS/kb) with or without 300 nM RfaH-NTD (**C**) or 50 nM GreB (**D**). Samples were removed at 0.33, 0.66, 1, 1.5, 2, 3, 4, 8, and 16 min after addition of NTPs and separated by denaturing PAGE.

The following figure supplements are available for figure 6:

**Figure supplement 1**. At 1 mM NTPs, NusG partially suppressed H-NS effects on pausing and GreA more significantly suppressed H-NS effects.

**Figure supplement 2**. Electrophoretic gel image showing that H-NS-induced RNAP backtracking was rescued by GreB.

**Figure supplement 3**. The H-NS stimulated C346 pause readily backtracked, whereas H-NS-resistant pausing at C370 occurred without obligate backtracking.

(*Figure 6—figure supplement 3H*). Thus, pausing at a sequence significantly enhanced by H-NS (C346) occurred significantly only when delayed and then readily backtracked to create a long-lived pause, whereas pausing at a sequence much less affected by H-NS (C370) exhibited much less potential for backtracking.

Taken together, these data suggest that H-NS promotes pausing by stimulating backtracking. Both the sequences of strongly H-NS-stimulated pauses and the stronger effects of GreB than RfaH-NTD are consistent with this hypothesis. The C370 pause that showed greater response to RfaH-NTD than either GreB or H-NS may reside mostly in the elemental pause state, entry to which may be suppressed by RfaH-NTD-induced forward translocation. This pause occurs at a relatively C-rich RNA:DNA hybrid, with both −10 G and −11 G present in the consensus pause sequence and is thus a strong candidate for a non-backtrack, elemental pause. Examination of the potential for backtracking at C346 and C370 sequences confirmed this interpretation.

## H-NS-stimulated pausing expands the kinetic window for Rho-dependent termination

Genomic locations of H-NS filaments and transcription termination caused by the Rho termination factor are highly correlated, raising the possibility that H-NS-stimulated pausing by aiding Rho-dependent termination (*Peters et al., 2012*). Deletion of *hns* is synthetically lethal with either chemical inhibition of Rho or mutations in *rho*, suggesting a functional role for H-NS in Rho-dependent termination (*Tran et al., 2011*; *Peters et al., 2012*). Rho is a homohexameric ATP-dependent RNA helicase that binds to ~80 nt of unstructured C-rich RNA (*Figure 7A*, inset) (reviewed in *Peters et al. (2011)*). Once bound to a nascent RNA transcript, Rho translocates 5′–3′ along the RNA until it reaches the EC, where it terminates transcription. Thus, the elongation rate of the EC determines a kinetic window during which it can be acted upon by Rho; increases in the frequency and strength of pausing can assist in termination by extending this kinetic window (*Jin et al., 1992*).

To test directly whether H-NS-stimulated pauses increased the kinetic window for Rho-dependent termination, we added Rho and 30 µM NTPs to halted ECs on DNAs with or without bridged H-NS filaments at 28°C, a temperature at which Rho terminated transcription and H-NS stimulated pausing in control experiments (not shown). To distinguish possible Rho stimulation of pausing from true Rho-dependent termination, we separated the RNAs after 32 min into an EC-bound fraction retained on beads via a His$_{10}$ tag on RNAP and a supernatant fraction containing the terminated RNAs (*Figure 7A,C*, *Figure 7—figure supplement 1A*). We observed significant termination only when Rho was present; bridged H-NS filaments alone did not cause RNA release (compare red and purple traces, *Figure 7A*). However, Rho terminated ECs at earlier positions on DNA in bridged filaments than DNA lacking H-NS (compare blue and purple traces, *Figure 7A,E*). Strikingly, bridged H-NS facilitated Rho termination at sites poorly utilized by Rho alone and at which H-NS strongly stimulated pausing (e.g., C346 and C393, *Figure 7—figure supplement 1A*, *Table 1*). The C346 and C393 pauses are strong candidates for backtrack pauses that were affected by both H-NS and GreB, whereas a site of Rho termination observed in both the presence and absence of H-NS (C370) is a candidate for a non-backtracked, elemental pause (*Table 1*, *Figure 7E*; see above). This result suggests that non-backtracked pauses may be better natural substrates for Rho than backtracked pauses (see 'Discussion'). However, strong H-NS stimulation of pausing may allow Rho termination at backtrack sites by increasing the kinetic window for Rho action at the sites as backtracked ECs are known to be poor substrates for Rho (*Dutta et al., 2008*). An alternative hypothesis is that direct H-NS–Rho interaction aids termination. However, H-NS in linear filaments did not synergize with Rho; the patterns of Rho termination on linear filaments vs DNA alone were almost identical (*Figure 8*, *Figure 8—figure supplement 1*). We conclude that H-NS-stimulated pausing on bridged filaments increases the kinetic window for Rho action (*Jin et al., 1992*), and the apparent preferential effect of Rho on backtrack pauses may allow Rho to terminate at otherwise suboptimal sites.

## H-NS and NusG synergistically aid in Rho-dependent termination

In addition to its pause-suppressing activity, NusG enhances Rho-dependent termination through direct interactions between Rho and NusG (*Figure 7A*, inset) (*Li et al., 1993*; *Pasman and von Hippel, 2000*; *Mooney et al., 2009*; *Chalissery et al., 2011*). In vivo, NusG improves the efficiency of Rho termination at a fraction (~20%) of terminators with suboptimal Rho binding sites (*Sullivan and Gottesman, 1992*; *Peters et al., 2012*). Consistent with in vivo effects of H-NS on Rho-dependent termination, genetic studies have shown that gain-of-function mutations in *hns* suppress loss of function mutations in *rho* and *nusG*, suggesting that all three function in the same pathway (*Saxena and Gowrishankar, 2011*). Further, H-NS filaments are found near NusG-dependent Rho terminators in vivo (*Peters et al., 2012*).

To test if NusG affects the synergy between Rho and H-NS, we compared the elongation and termination profiles of NusG and Rho to reactions containing NusG, Rho, and bridged H-NS (*Figure 7B,D*). We found that NusG-enhanced Rho termination at H-NS-dependent pause sites that are strong candidates for backtracking. The addition of NusG robustly aided in Rho-dependent termination, consistent with previous reports. During transcription of bridged H-NS filaments, NusG stimulated termination of the C346 and C393 pauses more than the C370 pause and allowed termination at the upstream U169 pause (*Figure 7B,E*, *Figure 7—figure supplement 1A*). NusG also altered the preferred positions of termination at the U588 pause. These effects were exacerbated when bridged H-NS was present, inhibiting overall elongation and increasing termination relative to samples

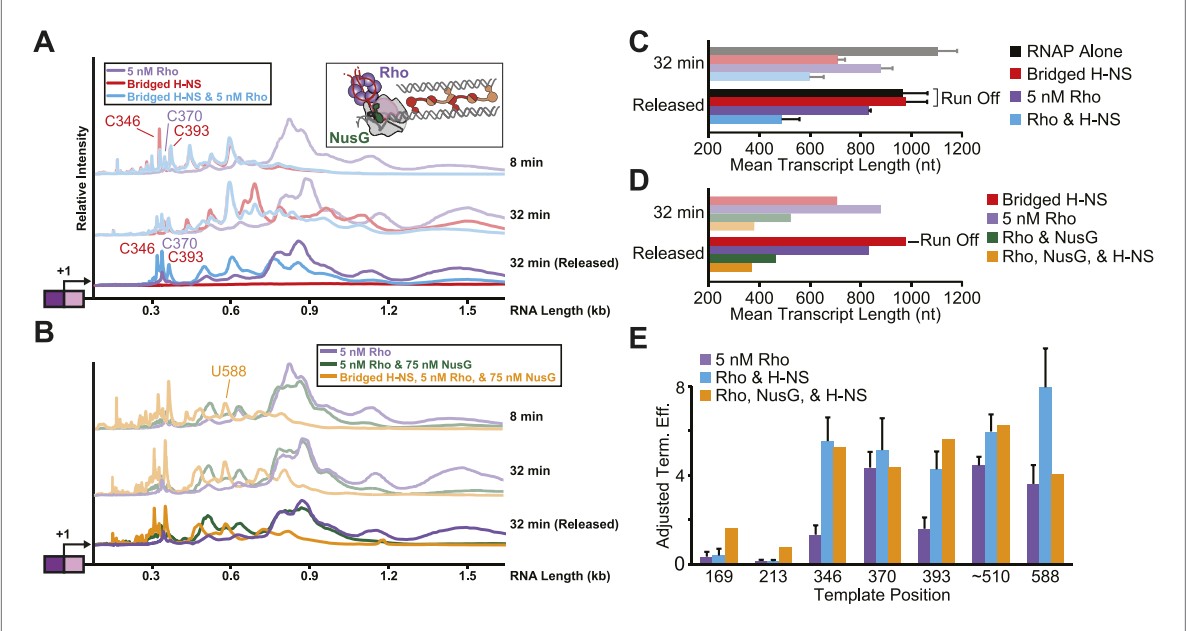

**Figure 7**. Stimulation of pausing by bridged H-NS filaments aided Rho-dependent termination. (**A**, **B**) Densitometry profiles of transcripts produced at 28°C, 8 mM $Mg^{2+}$, and 30 μM each NTP from the $\lambda P_R$-*bgl* template in bridged filaments (66 H-NS/kb) with or without 5 nM Rho (**A**) and with or without 75 nM NusG (**B**). Samples were removed at 2, 4, 8, 16, and 32 min after NTPs were added and separated by denaturing PAGE. To detect release of Rho-terminated transcripts, a 32-min sample was separated into released and EC-bound fractions using paramagnetic $Co^{2+}$ beads that bind the $His_{10}$-tagged RNAP. The released supernatant fraction was separated by denaturing PAGE and converted to densitometric profiles shown as darker colors. (**C**, **D**) Mean transcript lengths and standard deviations were calculated from at least two (**C**) or four (**D**) independent experiments. (**E**) Rho termination efficiencies were calculated as the fraction of released transcripts divided by the total transcripts, with averages and standard deviations from at least three independent experiments.

The following figure supplement is available for figure 7:

**Figure supplement 1**. Stimulation of pausing by bridged H-NS filaments aided Rho-dependent termination.

containing only Rho or both NusG and Rho. These results suggest that NusG, like H-NS, may aid Rho in terminating backtracked ECs. Synergy between the actions of NusG and H-NS at Rho terminators could play an important role in transcriptional silencing by H-NS in vivo (see 'Discussion').

## Discussion

Our results establish that the bacterial H-NS nucleoprotein filament, a central organizer of the bacterial nucleoid structure, can slow transcript elongation by RNAP and increase Rho-dependent termination by increasing transcriptional pausing at specific sites. The effects of H-NS on sequence-specific pausing resemble effects of eukaryotic nucleosomes on RNAPII elongation (*Studitsky et al., 1995*, *1997*; *Kireeva et al., 2005*; *Bintu et al., 2012*), but unlike the effects of nucleosomes, which are modulated by chromatin remodelers and histone tail modifications (*Bintu et al., 2012*; *Skene et al., 2014*), we found that changes in the bridging behavior of H-NS filaments govern its effects on elongating bacterial RNAP. These findings have several important implications for bacterial gene regulation.

### Mechanism of H-NS effects on pausing and termination

Bridged H-NS filaments slow RNAP at a subset of pause sites at which RNAP appears susceptible to backtracking, as evidenced by the large effect of GreB in suppressing H-NS stimulation of pausing at these sites, by the presence of short U-tracts just upstream from the pause sites at which H-NS has the largest effects, and by direct detection of backtracking for the C346 H-NS stimulated pause. The existence of both H-NS-sensitive and largely H-NS resistant pause sites, both of which occur at sequences conforming to the recently described consensus pause sequence (*Larson et al., 2014*), is consistent with the view that RNAP remains in a non-backtracked, elemental pause state at some sites

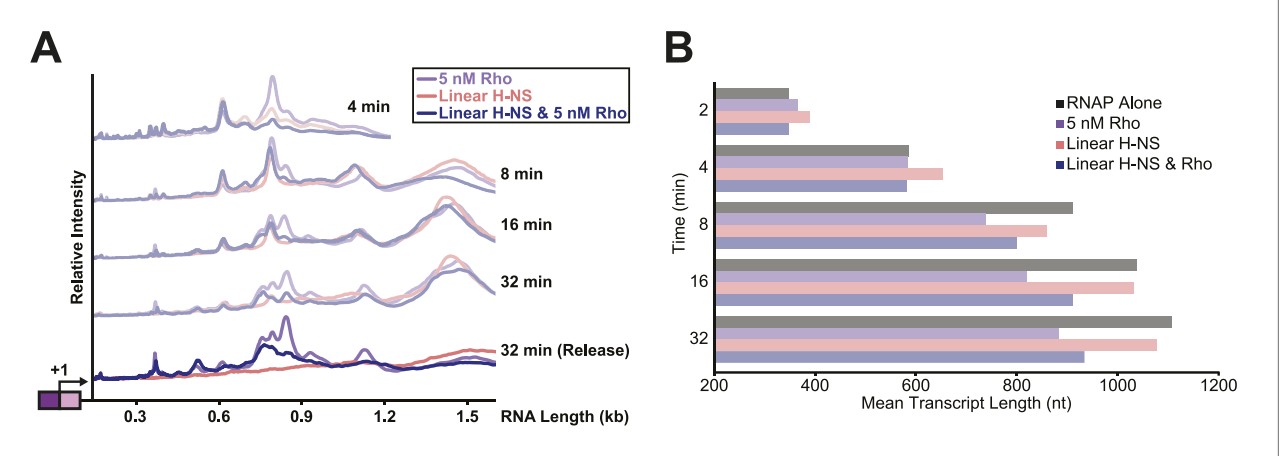

**Figure 8**. Linear H-NS filaments did not aid in Rho-dependent termination. (**A**) Densitometry profiles of transcripts produced at 28°C, 8 mM $Mg^{2+}$, and 30 μM each NTP from the $\lambda P_R$-*bgl* template in linear filaments (200 H-NS/kb) with or without 5 nM Rho. Samples were removed at 2, 4, 8, 16, and 32 min after addition of NTPs and separated by denaturing PAGE. Rho-terminated, released transcripts for the 32-min sample were determined as described in the legend to *Figure 7*. (**B**) Mean transcript lengths were averaged from two independent experiments.

The following figure supplement is available for figure 8:

**Figure supplement 1**. Linear H-NS Filaments did not aid in Rho termination.

and that such paused ECs are less susceptible to inhibition by H-NS. At least two hypotheses might explain the effects of H-NS at the class of sites more susceptible to backtracking: (1) bridged H-NS might create a physical barrier that blocks forward translocation and thus favors reverse translocation of RNAP (road-blocking model) or (2) the structure of the bridged filament entraps elongating RNAP in small, topologically fixed domains by binding both upstream and downstream of the EC (topological model; *Figure 9A*). Although we lack sufficient information to exclude either mechanism unambiguously and both could be contributory, our results favor the topological model.

If the road-blocking mechanism is correct, then H-NS must bind DNA significantly more tightly in the bridging mode than in the linear mode. The off-rate from DNA of H-NS in bridged filaments has been estimated to be 1.5 $s^{-1}$ at 20°C (*Dame et al., 2006*). However, we observed strong H-NS effects at RNAP pauses with much longer dwell times under similar conditions (pauses of minutes or more at 20°C and 30 μM NTPs) and where the average elongation rate of RNAP is ~1.3 $s^{-1}$. A priori, H-NS dissociates too fast to slow pause escape so dramatically. Although direct measures of H-NS exchange rates under our conditions will be needed to draw firm conclusions, the available data appear inconsistent with a road-blocking mechanism. Further, prior demonstrations of RNAP road-blocking used *lac* repressor or non-cleaving *Eco*RI proteins that bind with $K_d$s of 0.3–1 pM (~$10^4$ tighter than we observe for H-NS, 2 nM), whereas only partial road-blocking was observed with an *Eco*RI mutant exhibiting a $K_d$ of 30 pM (still 100× tighter than H-NS) (*Deuschle et al., 1986*; *Pavco and Steege, 1990*). These data lead us to favor the topological model.

A topologically constrained domain created by H-NS bridging around elongating RNAP will cause over-winding (positive supercoiling) of DNA downstream of an EC and under-winding (negative supercoiling) of DNA upstream of an EC by ~1 turn for every 10.3 rounds of nucleotide addition (*Figure 9A*) (*Liu and Wang, 1987*). These topological forces strongly inhibit translocation and favor backtracking of RNAP (*Ma et al., 2013*). In contrast, linear H-NS filaments will allow either the EC or the upstream and downstream DNA to freely rotate as transcript extension occurs. Thus, the topological model readily explains why the shift to linear filaments at higher H-NS concentrations reverses the pause-stimulating effects of H-NS, whereas the road-block model must invoke a large change in H-NS affinity that, while possible, lacks obvious physical rationale.

The preferential effect of H-NS on backtrack pauses also may explain the variable effects of Rho and NusG at different pause sites. Rho terminates non-backtracked ECs more readily than backtracked ECs (*Pasman and von Hippel, 2000*; *Dutta et al., 2008*) and NusG enhances the rate at which Rho dissociates ECs (*Burns and Richardson, 1995*). In the absence of H-NS, Rho terminated apparent

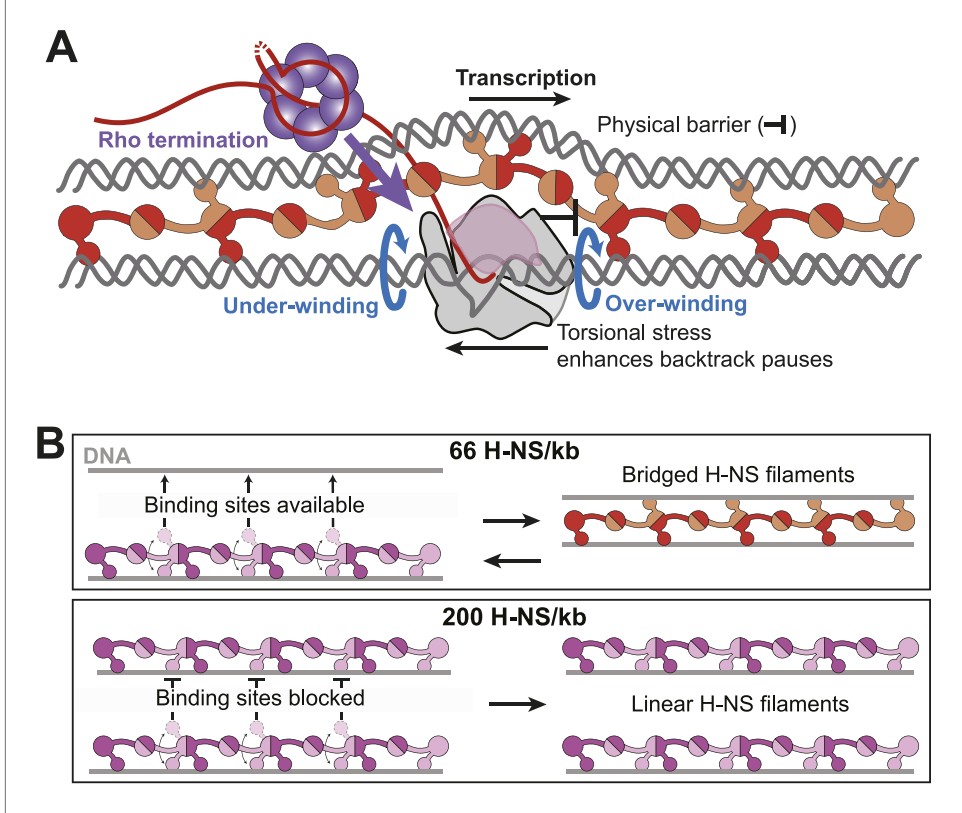

**Figure 9**. Models for H-NS effects on pausing, Rho termination, and DNA bridging. (**A**) As RNAP elongates through bridged filaments, pause durations increase for one or both of two reasons: (i) the off-rate of bridged H-NS is slower than the elongation rate of RNAP, leading to a roadblock (physical barrier; black bar); or (ii) H-NS bridging creates a closed topological domain that accumulates positive and negative supercoiling (torsional stress) in front and behind the EC, respectively, because free rotation of the DNA is blocked by bridged H-NS contacts and free rotation of the EC is blocked by steric clash between the bridged H-NS–DNA filament and the nascent RNA, including macromolecules like Rho or ribosomes bound to the nascent RNA (*Liu and Wang, 1987*). Blue arrows depict the rotation of DNA required to avoid torsional strain when the DNA is unconstrained. Both the under-winding (behind EC) and over-winding (in front of EC) torsional stresses will increase the propensity for RNAP to backtrack, thus increasing the duration of pauses that involve backtracking and increasing the kinetic window for Rho-dependent termination at backtrack pauses. (**B**) At 66 H-NS/kb, H-NS-free DNA segments allow DNA-binding domains from initially formed linear filaments to interact and form bridged filaments (top). At 200 H-NS/kb, all DNA segments become occupied by H-NS, leaving no available unbound DNA for formation of bridged filaments (bottom).

non-backtracked pauses more readily than apparent backtracked pauses. By increasing the time window for Rho action on backtrack pauses, H-NS may increase the sites at which Rho can dissociate a significant fraction of ECs. NusG, which exhibited a preferential effect on the apparent backtracked pauses, may be especially synergistic with H-NS because it both shifts ECs to less backtracked registers and directly aids EC dissociation by Rho, thus overcoming the apparently greater barrier for Rho dissociation at backtrack pause sites.

## Formation of bridged H-NS filaments is suppressed by high H-NS concentration

Previous studies have established that <5 mM $Mg^{2+}$ favors linear (stiffened) H-NS filaments and >5 mM $Mg^{2+}$ favors bridging (*Liu et al., 2010*), $Mg^{2+}$ levels within those estimated for the *E. coli* cytoplasm (1–10 mM free $Mg^{2+}$; ≥100 mM including bound $Mg^{2+}$) (*Moncany and Kellenberger, 1981*; *Cayley et al., 1991*). Our AFM observations support bridged-linear filament switching in the 1–10 mM $Mg^{2+}$ range, but it is currently unclear whether $Mg^{2+}$ binds H-NS, DNA, or both to favor the switch to bridging. Biophysical studies of H-NS are needed to elucidate this mechanism.

Our results add H-NS concentration as a second important parameter governing the bridged-linear switch. As H-NS levels saturate all available DNA binding sites, assuming a site size of ~5 bp/H-NS monomer, bridging becomes disfavored (*Figure 1*). The simplest explanation for this effect is that bridging requires the interaction of an H-NS-free DNA segment with an H-NS bound segment to nucleate formation of a 2-duplex bridged structure such as that proposed by *Arold et al. (2010)* (*Figure 9B*, *Figure 1—figure supplement 1*). At high H-NS levels, all DNA segments may become occupied by H-NS even if bound by only one of the two available DNA-binding domains at each CTD–CTD junction in the filament. At this point, linear filaments may become favored because no free DNA sites are sterically available, even if the intrinsic stability of the linear filament is less than that of the bridged filament structure.

Based on prior measurements (*Ali Azam et al., 1999*), H-NS is present in *E. coli* well below the saturating levels needed to produce linear filaments (equivalent to 2 H-NS/kb in exponentially growing cells containing ~2 genome equivalents of DNA). However, many additional factors including sequestration of DNA by other proteins, low $Mg^{2+}$ concentration, or elevated expression of H-NS could make the switch to linear filaments possible. If the topological model of H-NS action on ECs is correct, then one particularly interesting aspect of bridged-linear switching is that linear filaments might readily inhibit transcription initiation at promoters by interfering with RNAP binding with less effect on elongating RNAP, whereas bridged filaments might inhibit both initiation and elongation.

## Physiological consequences of H-NS effects on transcript elongation

The roles of H-NS nucleoprotein filaments in repressing transcription initiation by occluding RNAP or activator binding at promoters are well established, including repressing horizontally transferred genes, allowing expression of genes in pathogenicity islands during bacterial infections and suppression of pervasive antisense and non-coding transcription (*Dorman, 2007*; *Fang and Rimsky, 2008*; *Fass and Groisman, 2009*; *Singh et al., 2014*). Effects on elongating RNAP have been strongly indicated by frequent downstream locations of H-NS binding sites that affect gene expression and by H-NS synergy with Rho in vivo (*Dole et al., 2004*; *Saxena and Gowrishankar, 2011*; *Peters et al., 2012*). However, direct biochemical evidence has been lacking, and some results have questioned whether H-NS can affect elongating RNAP (*Lucht et al., 1994*; *Dame et al., 2006*). Our finding that bridged filaments enhance pausing and Rho-dependent termination in vitro provides a mechanistic basis for the in vivo effects of H-NS on transcript elongation and suggests several ways that changes in the effects of H-NS filaments can contribute to bacterial gene regulation.

Several factors known to affect the formation of bridged filaments or elongating RNAP are also known to vary among growth environments or conditions. Increased temperature upon bacterial invasion of a mammalian host is thought to play a key role in H-NS-mediated up-regulation of genes involved in pathogenesis or symbiotic growth (*Goransson et al., 1990*; *Amit et al., 2003*; *Trachman and Yasmin, 2004*; *Ono et al., 2005*; *Yang et al., 2005*; *Bouffartigues et al., 2007*). Our results suggest that H-NS bridging is inhibited at 37°C and that effects of H-NS on elongating RNAP mediated by bridged H-NS decrease above 30°C. Thus, an important component of H-NS-mediated temperature regulation of laterally transferred genes, pathogenesis genes, and transcriptionally silenced genes may be aided by decreased H-NS stimulation of Rho-dependent termination at temperatures encountered in mammalian hosts.

The concentration of H-NS is highest in exponential phase, and decreases by about a factor of four in stationary phase (*Ali Azam et al., 1999*); conversely, H-NS is induced by cold–shock (*La Teana et al., 1991*). Although the complexity of proteins interacting with DNA in cells precludes simple inferences, these changes in H-NS concentrations might influence levels of bridging and thus the magnitude of effects on elongating RNAP similar to the H-NS concentration effects we observed in vitro. Because small changes in effects on transcript termination are cumulative, unlike the switch-like effects that operate during initiation, modest changes in the amounts of bridging could be magnified (e.g., increasing termination from 1% to 10% at 20 sites in an operon will reduce expression by a factor of 7).

Changes in $Mg^{2+}$ levels, which strongly influence the propensity for bridging and the magnitude of effects on RNAP over relatively narrow changes in concentration (2 mM to 8 mM; *Figure 2*) are thought to decrease in response to external $Mg^{2+}$ levels during some bacterial infections (*Groisman, 1998*). Combined with increases in temperature, which decrease effects of H-NS on elongating RNAP and increase during infection, reduced $Mg^{2+}$ levels could increase gene expression by decreasing H-NS bridging.

Finally, multiple additional proteins that interact with H-NS could strongly influence the effects of H-NS on transcript elongation. For example, the H-NS paralog StpA may be present in filaments with H-NS (*Leonard et al., 2009*; *Uyar et al., 2009*) and could alter bridging potential (*Lim et al., 2012*). The *E. coli* Hha and YdgT proteins, although lacking DNA-binding domains, share the oligomerization fold of H-NS, may interact with H-NS filaments, and may modify their properties (*Saxena and Gowrishankar, 2011*; *Ali et al., 2013*; *Ueda et al., 2013*). Indeed our preliminary results suggest that both StpA and Hha-modified H-NS filaments more strongly inhibit transcript elongation by RNAP than bridged H-NS filaments alone (MK, BB, DH and RL, unpublished observations). Hha is thought to govern both initiation and elongation of the *hly* operon in uropathogenic *E. coli* (*Juarez et al., 2000*; *Nieto et al., 2000*; *Madrid et al., 2002*; *Ueda et al., 2013*) and intriguingly is regulated by RfaH, which we found modestly aided elongation through H-NS filaments.

Taken together, these results suggest multiple ways that inhibition of transcript elongation by bridged H-NS filaments may play crucial roles in bacterial gene regulation. Like the growing appreciation of the interplay between chromatin structure, promoter-proximal pausing, transcriptional regulation, and RNA processing in eukaryotes, these effects of H-NS on elongating bacterial RNAP are variable and may be modulated by cellular and environmental conditions. Much work remains to establish the mechanistic bases of the links between these effects and the repression of horizontally transferred genes, pathogenicity islands, and pervasive antisense and noncoding transcription. Our study provides a first step toward this mechanistic understanding.

## Materials and methods

### Materials

DNA oligonucleotides were obtained from Integrated DNA Technologies (Coralville, IA), $[\alpha\text{-}^{32}P]$NTPs and $[\gamma^{32}P]$ATP were from PerkinElmer Life Sciences (Waltham, MA), NTPs were from GE Healthcare Life Sciences (Piscataway, NJ), and 3'–deoxyNTPs were from Trilink (San Diego, CA).

### Plasmids and templates

The C-terminally His$_6$-tagged H-NS expression plasmid pET21-HNS-cHis6 was the kind gift of Sohail Akhtar and Aseem Ansari. Plasmid pMK110 encoding the *bgl* transcription template (*Figure 1*) was described previously (*Haft et al., 2014*). Plasmid pMK121 (*Figure 2—figure supplement 3*) was constructed by inserting a 1091-bp PCR product containing *bglF* from the *E. coli bgl* operon between the *Spe*I and *Pst*I sites of pIA267 (*Artsimovitch and Landick, 2002*). PCR was performed using *E. coli* chromosomal DNA, forward primer 5'-TCGAGCACTAGTCAGGCGATAACAAAGGGGTA, and reverse primer 5'-TATGCTCTGCAGGAATTCTGCGCAACGCGATTACGTT. After purification, the PCR product was digested with *Spe*I and *Pst*I and ligated into similarly cut pIA267. Plasmids pMK122, pMK124, and pMK126 (*Figure 2—figure supplement 1*) were constructed by inserting regions from 312, 707, and 912 nt downstream of the transcription start site of pMK110, respectively, into the *Spe*I site of pIA267 using forward primers 5'-GCATACTAGTTTAACGCTTCTTCCCCTAGC (pMK122), 5'-GCATACTAGTTT TTGGTGATTTGCATGTTCA (pMK124), and 5'-GCATACTAGTGACCATGCTCGCAGTTATT (pMK126) along with the reverse primer 5'-GCATACTAGTGGCGATGAGCTGGATAAACT. Transcription templates were generated by PCR amplification from pMK110, pMK121, pMK122, pMK124, and pMK126 using forward primer 5'-CGTTAAATCTATCACCGCAAGGG and reverse primer 5'-CAGTTCCCTAC TCTCGCATG. pMK110-520 template was generated from pMK110 with the same reverse primer used above and forward primers 5'-CACTAATTTATTCCATGTCACACTTTTCGCATCTTTTTTATGCT ATAATTATTTCATGTAGTAAAGAGGAATATGACTTAAGAGTTCGC or 5'-CACTAATTTATTCCATGT CACACTTTT. PCR products were electroeluted from a 1% agarose gel, phenol extracted, and ethanol precipitated.

### Proteins

RNAP (*Nayak et al., 2013*), σ$^{70}$ (*Gribskov and Burgess, 1983*), NusG (*Mooney et al., 2009*), RfaH-NTD (*Kolb et al., 2014*), GreA (*Feng et al., 1994*), GreB (*Feng et al., 1994*), and Rho (*Mooney et al., 2009*) were purified as described previously. H-NS was purified by overexpression in *E. coli* strain BL21 (λDE3) containing pET21-HNS-cHis6, and grown at 37°C in Luria broth supplemented with 0.1 mg Ampicillin/mL to an apparent OD$_{600}$ of 0.4. The temperature was lowered to 30°C, 500 μM isopropyl-1-thio-β-D-galactopyranoside was added, and the culture was grown for 4 hr with shaking. Cells were pelleted at 3000×*g* for 15 min at 4°C, resuspended in H-NS lysis buffer (20 mM Tris–HCl pH 7.5,

100 mM NaCl, 5% glycerol, 2 mM EDTA, 1 mM dithiothreitol [DTT], and 1 mM β-mercaptoethanol) supplemented with 0.1 mg PMSF/ml and a protease inhibitor mix (0.0125 mg of benzamide/ml, $2 \times 10^{-4}$ mg of chymostatin/ml, $2 \times 10^{-4}$ mg of leupeptin/ml, $4 \times 10^{-5}$ mg of pepstatin/ml, $4 \times 10^{-4}$ mg of aprotonin/ml, and $4 \times 10^{-4}$ mg of antipain/ml), and lysed by sonication. All the subsequent steps were performed at 4°C. The H-NS containing lysate was then enriched by mixing with polyethylenimine (PEI; avg. MW 60 K; Acros Organics) to 0.6%, incubating for 5 min, and then collecting the H-NS-containing precipitate at 11,000×*g* for 15 min. The PEI pellet was washed by gently resuspending in PEI wash buffer (10 mM Tris–HCl pH 7.5, 150 mM NaCl, 0.1 mM EDTA, 5% glycerol, and 1 mM DTT), followed by centrifugation at 11,000×*g* for 15 min. H-NS was eluted from the nucleic acid pellet by gently resuspending in PEI elution buffer (10 mM Tris–HCl pH 7.5, 600 mM NaCl, 0.1 mM EDTA, 5% glycerol, and 1 mM DTT) followed by centrifuging at 11,000×*g* for 15 min. The eluted supernatant was then precipitated by slow addition of ammonium sulfate with gentle stirring to 37%. The solution was stirred overnight, and the precipitate was collected by centrifugation at 27,000×*g* for 15 min. The H-NS precipitate was resuspended in 35 ml of buffer A (20 mM Tris–HCl pH 7.5, 500 mM NaCl, and 5 mM β-mercaptoethanol) containing 5 mM imidazole, loaded onto a 5 ml HisTrap column (GE Healthcare), washed with 30 ml of buffer A containing 5 mM imidazole, and eluted over a gradient of 5–500 mM imidazole over 10 min at a flow rate of 2 ml/min. H-NS containing fractions were combined and dialyzed against buffer B (10 mM Tris–HCl pH 7.5, 0.1 mM EDTA, 5% glycerol, 100 mM NaCl, and 1 mM DTT) overnight. The dialyzed H-NS was loaded onto a 5 ml HiTrap heparin column (GE healthcare), washed with 30 ml of buffer B, and eluted over a gradient of 0.1–0.9 M NaCl over 12 min. Fractions containing H-NS were pooled and dialyzed into H-NS storage buffer (20 mM Tris–HCl pH 7.5, 300 mM KCl, and 10% glycerol).

## In vitro transcription elongation assays

RNAP holoenzyme (core ββ′α$_2$ω plus σ$^{70}$) was prepared by incubating twofold molar excess of σ$^{70}$ with core for 30 min at 30°C in RNAP storage buffer (20 mM Tris–HCl pH 7.9, 100 mM NaCl, 10 mM MgCl$_2$, 0.1 mM EDTA, 1 mM DTT, and 40% glycerol). Halted ECs were formed by incubating 10 nM λP$_R$-*bgl* template (linear pMK110 template) and 15 nM RNAP holoenzyme in EMSA buffer (*Stonehouse et al., 2011*) (40 mM HEPES-KOH pH 8.0, 100 mM potassium glutamate, 0.022% NP-40, 100 µg bovine serum albumin/ml, and 10% glycerol supplemented with 8 or 2 mM magnesium aspartate) and combined with 150 µM ApU, 10 µM ATP and UTP, 2.5 µM GTP, and 0.37 µM (10 µCi) [α-$^{32}$P]GTP for 10 min at 37°C to trap A26 ECs. H-NS (1 or 3 µM ; 66 H-NS/kb or 200 H-NS/kb. respectively) was added to the A26 ECs and incubated for 20 min at 20°C. When specified in the figure legends, 75 nM NusG, 5 nM Rho, 50 nM GreB, 500 nM GreA, or 300 nM RfaH-NTD was added. Transcription was then allowed to resume by adding ATP, UTP, CTP, and GTP (30 µM each), 100 µg rifampicin/ml, and 0.1 U RNasin/µl (Promega, Madison, WI). Samples (10 µl) were taken at indicated time points by mixing with EDTA (20 mM final), immediately phenol:cholorfom extracted, and ethanol precipitated. Reaction pellets were resuspended in Formamide Running Dye (95% formamide, 15 mM EDTA pH 8, 0.05% bromophenol blue, and 0.05% xylene cyanol). Samples were heated for 2 min at 90°C and separated by electrophoresis in a denaturing 6% or 12% polyacrylamide gel (19:1 acrylamide:bisacrylamide) in 1.25 mM Na$_2$EDTA, and 44 mM Tris borate, pH 8.3 (0.5× TBE) plus 7 M urea. Gels were exposed to a PhosphorImager screen, scanned using a Typhoon PhosphorImager, and quantified using ImageQuant software (GE Healthcare, Waukesha, WI).

Densitometry profiles and mean transcript lengths were generated as previously described (Haft et al. PNAS 2014). ECs in reactions chased with 1 mM NTPs were formed in EMSA buffer supplemented with 12 mM magnesium aspartate. All transcription experiments were repeated at least twice.

## Pause site mapping

Halted A26 complexes were formed on the λP$_R$-*bgl* template with 10 µM ATP and UTP and 2.5 µM GTP supplemented with 0.37 µM (10 µCi) [α-$^{32}$P] GTP. Transcription was restarted with 30 µM ATP, UTP, GTP, and CTP in the presence of 100 µg rifampacin/ml. Samples were collected and stopped as above (where H-NS was present) or by addition to 2× urea stop dye (10 M urea, 30 mM Na$_2$EDTA, 0.05% each of bromophenol blue, and xylene cyanol) and then separated by 8% denaturing PAGE (0.5× TBE plus 7 M urea). RNA ladders were generated by adding 10–50 µM of one 3′-deoxyNTP (G, A, U, or C) and 4 NTPs (30 µM each) to 10 nM halted ECs and incubating for 10 min before stopping by addition to 2× urea stop dye. RNA ladders were run alongside time points. Gels were visualized as described above.

## In vitro pause assay on scaffolds

Nucleic-acid scaffolds to reconstitute elongation complexes were assembled as described previously (*Kyzer et al., 2007*) using sequences shown in *Figure 2—figure supplement 1*. Briefly, (1 µM RNA) and template DNA (2 µM) were annealed in reconstitution buffer (10 mM Tris–HCl, pH 7.9, 40 mM KCl, and 5 mM $MgCl_2$). Scaffolds were diluted 10-fold with elongation buffer (EB; 25 mM HEPES-KOH, pH 8.0, 50 mM KCl, 5 mM $MgCl_2$, 1 mM DTT, 5% glycerol, and 25 µg acetylated bovine serum albumin/ml) and incubated with 0.5 µM RNAP for 10 min at 37°C. 300 nM non-template DNA (300 nM) was then added and incubation continues for 10 min at 37°C to form halted ECs. The RNA was extended one nucleotide with [α-$^{32}$P]CTP to form C346 scaffold ($EC_{C342}$) or C370 scaffold ($EC_{C368}$). On-the-fly transcription complexes were diluted 20-fold, and 10 µM CTP, UTP, and GTP ($EC_{C346}$) or CTP, ATP, and GTP ($EC_{C370}$) were added at either 37°C or 20°C. Samples were quenched by addition of an equal volume of 2× stop dye (10 M urea, 50 mM EDTA, 90 mM Tris-borate buffer, pH 8.3, 0.02% bromophenol blue, and 0.02% xylene cyanol) at 10, 20, 40, 60, 90, 120, 150, and 180 s at 37°C, or 20, 40, 60, 90, 120, 150, 180, and 360 s at 20°C. RNA was resolved by electrophoresis through a 15% denaturing polyacrylamide gel (0.5× TBE plus 7 M urea). For delayed transcription from $EC_{C342}$, CTP and UTP (10 µM each) were added at 37°C for 5 min to halt complexes at $EC_{C346}$. GTP (10 µM) was added, and samples were collected as described above. Delayed $EC_{C370}$s were walked to the pause by binding the His$_{10}$-tagged RNAP to $Co^{2+}$ magnetic beads during the last 5 min of incorporation labeling with [α-$^{32}$P]CTP. The bead-bound ECs were washed five times with 1 ml of EB and extended one nucleotide with 10 µM ATP. The complexes were washed again and extended to the pause site by addition of 10 µM CTP. Subsequently, CTP, ATP, and GTP (10 µM each) were added, and samples were collected as described above. For intrinsic or GreB-mediated cleavage assays, $EC_{C346}$ or $EC_{C370}$ complexes were elongated to the pause as described above. ECs were washed with 5× 1 ml of EB. Complexes were resuspended in cleavage buffer (25 mM Tris–HCl, pH 9.0, 50 mM KCl, 20 mM $MgCl_2$, 1 mM DTT, 5% glycerol, and 25 µg acetylated bovine serum albumin/ml) to induce intrinsic cleavage or were suspended in EB with or without 50 nM GreB to assay GreB-mediated hydrolysis. Samples were collected as described above and separated by 20% denaturing PAGE (0.5× TBE plus 7 M urea). Gels were exposed to a PhosphorImager screen, scanned using a Typhoon PhosphorImager, and quantified using ImageQuant software (GE Healthcare).

## Electrophoretic mobility shift assay

5'-$^{32}$P-labeled linear pMK110 fragment (10 nM or 10 pM; generated by PCR as described above) was incubated in EMSA buffer supplemented with 2 or 8 mM magnesium aspartate with various amounts of H-NS and incubated at 20°C for 20 min. Reactions were then resolved by electrophoresis though a 3% native PA gel (19:1 acrylamide:bisacrylamide) cast and run in 0.5× TBE plus 2.5% glycerol for 5 hr at 4°C, 20°C, or 37°C. The resulting polyacryamide gel was dried, exposed to PhosphorImager screen, and scanned using a Typhoon PhosphorImager.

## Atomic force microscopy

H-NS-DNA protein complexes were adsorbed onto APS-mica. Mica was functionalized with APS as described previously (*Shlyakhtenko et al., 2013*). 166.7 µM APS solution was incubated with mica for 30 min at room temperature. APS-mica was washed with water, dried under a stream of argon and then cured overnight and stored under argon at reduced pressure. 10 nM or 2 nM pMK110 template only or A26 ECs (described above) were incubated with various amounts of H-NS in AFM buffer (40 mM HEPES-KOH pH 8.0 and 100 mM potassium glutamate) supplemented with either 2 or 8 mM magnesium aspartate for 20 min at 20°C. A fraction of the H-NS complexes was loaded onto 3% native PAGE (described above). A fraction of complexes was directly applied to the APS-mica surface, incubated for 2 min at room temperature, washed with 600 µl water, dried under a stream of argon, and cured overnight under vacuum. Another fraction of the complexes was incubated on ice for 2 min and diluted 1:4 in ice-cold AFM buffer. The diluted complexes were deposited onto APS-mica at 4°C, incubated for 2 min at 4°C, washed with 600 µl ice-cold water, dried under a stream of argon, and cured overnight under vacuum. Dilution of the H-NS-DNA complexes was performed to lower H-NS background and aid identification of complexes in the higher undiluted samples. The remaining H-NS-ECs were used in transcription assays conducted in parallel (as described above). Images of samples were obtained with a MultiMode AFM (Digital Instruments NanoScope IV; now Bruker, Santa Barbara, CA) in tapping mode in air using TESPA-V2 cantilevers (Bruker). Images were analyzed for changes in

topology, changes in complex height and width with Gwyddion software. Additional contour length and persistence length measurements were made with Femtoscan software which uses previously established algorithms (*Rivetti et al., 1996*).

## Acknowledgements

We thank Michael Bellecourt, Erik Jessen, other members of the Landick laboratory, and Jason Peters for helpful discussions and comments on the manuscript. We also thank the University of Wisconsin Material Sciences Center (College of Engineering) for use of their AFM facilities. This work was supported by NIH grants F32 GM098009 to MVK and R01 GM38660 to RL.

## Additional information

### Funding

| Funder | Grant reference number | Author |
| --- | --- | --- |
| National Institutes of Health | R01 GM38660 | Robert Landick |
| National Institutes of Health | F32 GM098009 | Matthew V Kotlajich |

The funders had no role in study design, data collection and interpretation, or the decision to submit the work for publication.

### Author contributions

MVK, Conception and design, Acquisition of data, Analysis and interpretation of data, Drafting or revising the article, Contributed unpublished essential data or reagents; DRH, Acquisition of data, Analysis and interpretation of data; BAB, Conception and design, Acquisition of data, Analysis and interpretation of data; ZS, YLL, Analysis and interpretation of data, Drafting or revising the article; RL, Conception and design, Analysis and interpretation of data, Drafting or revising the article, Contributed unpublished essential data or reagents

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
