## [Decision Letter]

Thank you for sending your work entitled “Bridged nucleoprotein filaments of bacterial H–NS slow elongating RNA polymerase and aid Rho–dependent termination” for consideration at *eLife*. Your article has been favorably evaluated by James Manley (Senior editor), Nick Proudfoot (Reviewing editor), and 2 reviewers.

The Reviewing editor and the reviewers discussed their comments before we reached this decision, and the Reviewing editor has assembled the following comments to help you prepare a revised submission.

Your interesting paper has been reviewed by three experts in the field. They find your study worthy of *eLife* publication, provided you can address the issues that they have agreed on below:

1) This study is very much a mechanistic in vitro analysis of how H–NS influences bacterial transcription. However, it is important to validate these biochemical results by matching in vivo experiments ideally also showing the generality of your mechanistic observations. In particular, the observed fact that H–NS effects on transcription decrease at temperatures above 30°C would suggest that, in normally growing *E. coli* (which lives in the guts of warm–blooded organisms., i.e. at temperatures higher than 30°C), H–NS effects on transcription might not be significant. Also, are the observed windows of H–NS and Mg concentrations required for the elongation effects in vivo levels? To this end we think that in vivo analysis is required to support relevance of these in vitro data. A simple experiment would be to measure the rate of transcription of the sequence used this study or similar in living cells (by dot blot hybridization with early and late probes) as in Vogel and Jensen, 1994, J.Bact: 2807–2813. This method also reports on the success of reaching a later probe by RNAP, i.e. on irreversible arrests or premature termination. The method works not only with plasmids but also on the genome (Yuzenkova et al., 2014, NAR, 42:10,987–99), so should be easily adaptable for these purposes. It would be important to measure, at least, the rate of transcription in WT vs Δ*hns* strains at 20^°^C versus 37^°^C.

2) The composition of the EMSA running buffer should be described. Does it contain Mg^2+^ (or EDTA)? If not, then why do the authors assume that the filaments are unaffected during electrophoresis? To discriminate if higher temperature influences bridged filament formation, EMSA as in Figure 1 should be done at 37°C.

3) Distinguishing interwound and linear molecules as in Figure 1 is hard for the non expert. AFM images should be shown that compare the type of H–NS complexes observed between 2 and 8 mM of Mg (at 66 H–NS/Kb), which cause the significant shift in the formation of bridged complex at higher Mg concentration. The whole idea of the AFM experiments is to show that the 2 complexes seen in EMSA represent linear and bridge complexes. Are there known mutations in the H–NS protein (Head and Tail) that are known to perturb oligomerization? If so, such protein variants could be usefully used in the gel shift assays to test the authors' hypothesis.

4) The topological model proposed in Figure 9 suggests a potential mechanism by which H–NS bridging filaments entraps elongating RNAP and prevent forward movement. This model might also suggest that H–NS bridging promotes topological constraints behind the polymerase, which would be expected to reduce backtracking by multiple nucleotides. Yet, the data presented suggests that H–NS promotes RNAP pausing by stimulating backtracking. The ability of H–NS to increase backtracking needs to be confirmed more thoroughly. Immobilization of the reactions on beads (as in the Rho termination experiment) would allow isolation of paused ECs (as in Figure 2, 16 min lane). Washing away NTPs and then analysis of the susceptibility of paused ECs to Gre (A and/or B) should be carried out. A fast disappearance of pause bands would unambiguously show that the corresponding complexes have backtracked.

5) Figure 5 needs an artist's impression of how the EM data shows bridging on either side of EC.

---

## [Author Response]

*1) This study is very much a mechanistic* in vitro *analysis of how H–NS influences bacterial transcription. However, it is important to validate these biochemical results by matching* in vivo *experiments ideally also showing the generality of your mechanistic observations. In particular, the observed fact that H–NS effects on transcription decrease at temperatures above 30*^*°*^*C would suggest that, in normally growing* E. coli *(which lives in the guts of warm–blooded organisms., i.e. at temperatures higher than 30*^*°*^*C), H–NS effects on transcription might not be significant. Also, are the observed windows of H–NS and Mg concentrations required for the elongation effects* in vivo *levels? To this end we think that* in vivo *analysis is required to support relevance of these* in vitro *data. A simple experiment would be to measure the rate of transcription of the sequence used this study or similar in living cells (by dot blot hybridization with early and late probes) as in Vogel and Jensen, 1994, J.Bact: 2807–2813. This method also reports on the success of reaching a later probe by RNAP, i.e. on irreversible arrests or premature termination. The method works not only with plasmids but also on the genome (Yuzenkova et al., 2014, NAR, 42:10,987–99), so should be easily adaptable for these purposes. It would be important to measure, at least, the rate of transcription in WT vs Δ*hns *strains at 20*^*°*^*C versus 37*^*°*^*C.*

Because multiple points were contained in each enumerated paragraph in the aggregated review, we have numbered the responses 1a, 1b, etc to respond to each point explicitly. We describe 1b first, because the key issue of temperature dependent effects of H–NS in vivo is addressed there.

1b) “… the fact that H–NS effects on transcription decrease at temperatures above 30°C would suggest that, in normally growing *E. coli* (which lives in the guts of warm-blooded organisms; i.e. at temperatures above 30°C), H–NS effects on transcription might not be significant.”

Enteric bacteria, including *E. coli*, live both inside and outside the gut of warm-blooded mammals, and H–NS is already very well established as a major player in temperature-regulation of genes involved in pathogenesis (Falconi et al., 1998 EMBO J, 17: 7033; White-Ziegler et al., 1998, Mol Microbiol, 28: 1121), environmental responses (White-Ziegler et al., 2000, J Bacteriol, 182: 6391), and horizontal transfer (Mourino et al., 1994, Microb Pathog, 16: 249). It is estimated that ∼50% of *E. coli* on earth is within hosts and 50% is in non-host environments (Winfield and Groisman, 2003, Applied Environ Micro, 69:3687; Savageau, 1983, Am Nat, 122:732), where cells can thrive at 28–30°C. Indeed, the shift from non-host to host temperatures is thought to be a major component of regulatory program controlling symbiosis and pathogenesis in enteric bacteria ([28], Nature, 344: 682; Tagkopoulos et al., 2008, Science, 320: 1313). Thus, we disagree with the idea that reduced H–NS effect upon shift from ambient to body temperatures may not be significant. Rather, this shift in the properties of H–NS is already well established (e.g., [4], Biophys J, 84: 2467) and its role in genetic regulatory programs in *E. coli* is already well established (e.g., [70], Biochem J, 391: 203). Our results add a crucial mechanistic understanding to the basis for these temperature-dependent effects of H–NS. We have revised the Introduction to highlight this point by adding the sentence: “In vivo assays also establish that the H–NS block to transcription is greater in enterobacteria growing at 20–30°C outside hosts than at 37°C typical for symbiotic or pathogenic growth, and implicate H–NS in switching gene expression upon host invasion (28; 88; 70; 96).” We also now highlight this point in the Discussion, by adding a paragraph on temperature regulation.

1a) “… it is important to validate these biochemical results by matching in vivo experiments ideally showing the generality of your mechanistic observations.”

We appreciate and share the reviewers’ concern that our biochemical results should probe a mechanism relevant in vivo. We were surprised by this request for multiple reasons. First, we employed biochemical methods to dissect H–NS effects on transcript elongation and Rho-dependent termination in vitro because these temperature-dependent in vivo effects are already well established by an ample body of literature (e.g., [20]; [21]; [74]; [77]; [70]; [96]; [88], all cited in the revised manuscript). However, the complexity of multiple interacting regulators that govern these effects in cells make in vitro biochemical methods vastly more powerful to elucidate the underlying mechanisms. H–NS filaments inhibit transcript elongation in vivo presumably by aiding Rho-dependent termination, including in the bgl operon that we use as a test platform ([21]; [77]; [74]; Chandraprakash et al., 2013). We may have failed to establish this key point clearly in the Introduction. We have revised the Introduction and Discussion to emphasize the pre-existing knowledge about effects of H–NS on transcript elongation in vivo when we undertook this study (see point 1b). We think that it inappropriately discounts the enormous power of carefully designed biochemical experiments to request that they all be matched by parallel in vivo experiments attempting to show the same points.

Second, we found the request to conduct and include “matching in vivo experiments” surprising given the published *eLife* policies that new experiments will only be requested when they are essential to support the major conclusions and that requested experiments must be feasible in a reasonable time frame. A complete set of matching in vivo experiments would greatly lengthen the manuscript and could involve a lengthy investigation. Especially given the well-established in vivo precedence and the lack of precision possible with in vivo H–NS experiments (see below), we question the need to test every point established with biochemical experiments with parallel in vivo experiments in the context of a single manuscript. Additional in vivo studies would, of course, be desirable.

Third, although we are experienced in the Vogel and Jensen in vivo elongation rate assay recommended by the reviewers (see Ederth et al., 2006, JMB, 356:1163) and considered attempting such experiments during our study, we had decided against this approach due to major experimental complications predicted to make such experiments more problematic than perhaps envisioned by the reviewers. To conduct the minimal experiment requested by the reviewers, we would need to develop an assay system in which a strong H–NS filament-forming DNA was placed between an IPTG-inducible promoter and the *lacZ* gene on an assay plasmid, and then measure transcript elongation rates through it after induction of the promoter in wild-type and Δ*hns* cells at 37°C and 20°C.

Among the many problems, H–NS filaments are modulated in vivo by additional protein components, the best known of which are the H–NS paralogs StpA Hha, and YdgT. StpA can substitute for H–NS upon *hns* deletion. Hha and YdgT modify the properties of filaments in ways that are not currently understood. *hns* mutants are exceptionally sick, accumulate suppressors rapidly, and are cold-sensitive (i.e., the grow poorly at 20°C; Dersch et al., 1994 MGG 245:255). *hns stpA* double mutants exhibit even stronger phenotypes. These complex genetics and phenotypes mean that we cannot simply study an *hns* deletion. Rather, we would need to remove hns in a strain already containing *stpA* and *hha* deletions and the assay plasmid, and then perform the assay before suppressors accumulate. This need to P1 transduce Δ*hns* last also precludes use of *recA* strains, which typically are used to avoid issues with plasmid recombination. Without first sorting out the complex interactions of H–NS, StpA, Hha, and YdgT in vivo, we thought the in vitro approaches were far more promising. Using them, we can study proteins individually or in combination without worrying about H–NS paralogs, unknown regulators, suppressors, and poor cell viability.

Additionally, deletion of *hns* is known to cause cryptic promoters within an H–NS filament to become active ([81], Genes Dev, 28: 214). Thus, even with a strong, IPTG-inducible promoter placed upstream of an H–NS filament generating DNA segment, it is highly likely that upon *hns* deletion cryptic transcription from within the segment would confound measurement of elongating transcripts beginning at the inducible promoter. We were concerned that such cryptic initiation would make the in vivo experiments impossible.

Additionally, to make complete the requested experiment we must be able to measure elongation rates when H–NS filaments form on the bgl DNA, including when they are stabilized at 20°C. However, these filaments are highly likely to extend over an adjacent IPTG-inducible promoter and prevent adequate signal from being generated when H–NS is present. This is one of the major reasons we developed an in vitro system: the ability to form halted elongation complexes before H–NS is added allows us to circumvent inhibition of initiation by H–NS. We emphasize this point in the revised manuscript.

Additionally, all existing measurements of in vivo elongation rates using the Vogel and Jensen assay rely on genes in which transcription is coupled to translation, but H–NS–Rho cooperation is most prominent in untranslated and antisense transcripts (like the bgl antisense transcript used in our study; [74], Genes Dev, 26: 2621). Without translation, Rho rapidly terminates transcription in vivo and the only feasible way to conduct the requested experiment is to inhibit Rho with bicyclomcin (see Ederth et al., 2006). However, bicyclomycin is no longer freely available from the manufacturer (Astellas Pharmaceuticals) and is sold only by Santa Cruz for $315/mg. We estimate the minimal experiment requested by the reviewer (+/-*hns* at 20°C and 37°C) would cost $10-20K in bicylomycin, if done in triplicate. We contacted Santa Cruz about obtaining the necessary quantities, and were told it would require at least 2 months lead-time.

Despite these obstacles, we attempted to undertake the minimal experiment. We constructed plasmids containing the ∼1 kb bgl DNA inserted between P_T7A1lacO3,4_ and *lacZ* in the equivalent of pUV12 (see Ederth et al., 2006; Vogel and Jensen, 1994, J Bacteriol, 176: 2807) and knocked out the two known promoters (P1_bglG_ and P2_bglG_) that we could predict would interfere with the experiment upon *hns* deletion (although we expected cryptic promoters would still compromise the experiment). However, we found that Δ*stpA* Δ*hha* strains containing the plasmid could not readily transduced to Δ*hns* even at 37°C. Thus, we were unable to complete even the initial version of the minimal experiment in the time available.

For these reasons, we have opted to respond to the reviewers request for in vivo experiments by emphasizing the already extensive in vivo data that forms a precedent for our studies.

1c) “…are the observed windows of H–NS and Mg concentrations required for elongation effects in vivo levels?”

The answer to both questions is yes, although precise estimates of the effective concentrations of H–NS and Mg in vivo are difficult owing to occlusion of some DNA targets by other DNA binding proteins. In fact, we explicitly described both the in vivo Mg concentration (1–10 mM) and in vivo H–NS concentrations (2 H–NS /kb) in the Discussion section of the original submission, as well as the caveats that including bound Mg raises the level to 100 mM and considering occlusion by other DNA binding proteins could raise the effective H–NS concentrations (H–NS bound /kb) to levels easily within the ranges at which we see effects of H–NS on transcript elongation in vitro.

1d) “It would be important to measure, at least, the rate of transcription in WT vs. Δ*hns* strains at 20°C and 37°C” in vivo.

We appreciate why the reviewers believe this would improve the manuscript, but as we explain above the temperature-dependent effect of H–NS on elongation already is fairly well established from diverse, published experiments, and the experiment requested is more problematic than the reviewers likely appreciate. Further, the key effect of H–NS on gene regulation is to increase pausing in ways that cause increased Rho-dependent termination, which may or may not be reflected in a measurement of overall elongation rate measured by the Vogel and Jensen assay. Increased Rho-dependent termination could reflect increases in pausing at a small subset of locations such that the contribution to overall elongation rate might not be as significant as the reviewers expect, even though the effects on Rho-dependent termination at specific sites could be large. We agree that such measurements would be interesting because they could help determine whether effects on a limited set of pauses or larger effects on elongation rate occur, but for the reasons explained above (see 1a) we were unable to complete such experiments in the time frame available. Instead, we have provided an analysis of relative levels of pausing in DNA containing H–NS and not containing H–NS in cells that is consistent with the in vitro effects we report.

*2) The composition of the EMSA running buffer should be described. Does it contain Mg*^*2+*^
*(or EDTA)? If not, then why do the authors assume that the filaments are unaffected during electrophoresis? To discriminate if higher temperature influences bridged filament formation, EMSA as in*
Figure 1
*should be done at 37*^*°*^*C*.

We have added a complete description of the EMSA buffer and conditions to the Methods section. The buffer contains 1.25 mM EDTA and not Mg^+2^ (ninth paragraph of the Materials and methods section). In our experience, addition of Mg^2+^ to EMSA buffers compromises the electrophoresis because Mg^2+^ is an efficient charge carrier. We rely principally on the AFM results, not the EMSA, to assign the bridged filament; the fact that the shift in gel mobility occurs at H–NS concentrations that correspond to the shift from bridged to linear filaments in AFM strongly suggests the changed gel mobility reflects the bridged to linear transition detected by AFM, although other explanations could be possible. The reviewers have not suggested an alternative explanation. We hypothesize that the bridged filaments are stabilized by caging effects in the gel and by the low temperature once the Mg^2+^ is removed by electrophoresis. Such caging affects are known to stabilize protein-nucleic acid complexes during EMSA (Buratowski and Chodosh, 2001, Curr Protoc Mol Biol, 12.2; Fried & Liu, 1994, NAR 22:5054).

At the reviewers’ suggestion, we compared electrophoresis at 37°C (and 20°C) to results at 4°C. As the reviewers predicted, the apparent bridged filaments, which had a mobility relative to free DNA (R_f_) of 0.41, were destabilized at 37°C. Partial filaments formed below 0.8 µM H–NS were destabilized such that little of no smearing was evident relative to free DNA (slight smearing was evident when the filaments were formed at 20°C before loading onto the gel pre-equilibrated to 37°C). At very high H–NS concentrations and 37°C, slower migrating bands were evident—these shifts at high concentration may reflect some bridging at non-physiologically relevant H–NS concentrations and 37°C or some other H–NS binding phenomenon. The distinct 4°C EMSA pattern (bridged shifting to linear between 1 and 3 µM H–NS, corresponding to the shift seen in AFM images) and 37°C EMSA pattern (both 1 µM H–NS and 3 µM H–NS migrating at the linear filament R_f_) were seen whether filaments were formed at 20°C (as done in all our other experiments) or at 37°C. Thus, filaments appear to rearrange quickly in the gel wells upon loading. We have added this new result as Figure 1—figure supplement 3 and describe the new results in the manuscript (sixth paragraph of the Results section). We thank the reviewers for suggesting this experiment, which we believe strengthens our findings.

*3) Distinguishing interwound and linear molecules as in*
Figure 1
*is hard for the non expert. AFM images should be shown that compare the type of H–NS complexes observed between 2 and 8 mM of Mg (at 66 H–NS/Kb), which cause the significant shift in the formation of bridged complex at higher Mg concentration. The whole idea of the AFM experiments is to show that the 2 complexes seen in EMSA represent linear and bridge complexes. Are there known mutations in the H–NS protein (Head and Tail) that are known to perturb oligomerization? If so, such protein variants could be usefully used in the gel shift assays to test the authors’ hypothesis*.

We agree and have rearranged and relabeled the panels to make it easier to distinguish the difference. The 66 H–NS/kb panels at 2 and 8 mM Mg are now side-by-side in Figure 1 and more clearly labeled. The differences are most easily seen in the pseudo-3D plots, which we have moved to Figure 1 and placed as panel D directly below the top-down views in panel 1C. Note the pseudo-3D views are of DNA and filaments lacking ECs because the ECs would distort the scaling. We explicitly described the panels in the Results section so readers will more easily see the differences.

A variety of H–NS mutants have been isolated and studied in vivo (for a relatively complete list, see [63], H–NS as a defence system in Bacterial Chromatin ed. Dame and Dorman, Springer, pp. 251-322), but their effects on bridged vs. linear filaments has not been reported. Some mutants are known to lose or drastically reduce DNA binding, but such mutants would likely not be useful for the experiments suggested by the reviewers. To conduct such experiments, it would be necessary to purify the mutant H–NS proteins and characterize their effects on bridging using AFM before using them to test hypotheses about the EMSA assays. H–NS mutants are notoriously difficult to purify because even the wild-type protein is aggregation-prone. These studies could be undertaken, and we agree they would be worthwhile and likely to yield interesting results, but we believe they are beyond the scope of our present manuscript and would not possible to complete in the 2-month revision time frame.

*4a) The topological model proposed in*
Figure 9
*suggests a potential mechanism by which H–NS bridging filaments entraps elongating RNAP and prevent forward movement. This model might also suggest that H–NS bridging promotes topological constraints behind the polymerase, which would be expected to reduce backtracking by multiple nucleotides. Yet, the data presented suggests that H–NS promotes RNAP pausing by stimulating backtracking*.

We are confused by this statement. Topological constraints both in front and behind RNAP disfavor forward translocation and favor backtracking as transcription proceeds and the DNA is unable to rotate relative to RNAP. In front, the DNA becomes more positively supercoiled, which resists forward movement and promotes backtracking. In back, the DNA becomes more negatively supercoiled which favors backtracking and disfavors forward translocation. This phenomenon, known as the twin supercoiled domain model, is well-documented (e.g., see [57], Science, 340:1580) and reflects the fact that the DNA duplex must rotate to pass though RNAP. The model in Figure 9 depicts these effects with the circular blue arrows.

It is possible that the reviewers are referring to potential steric effects of bridged filaments. In front, collision of RNAP with H–NS would favor backtracking, but behind collision would disfavor backtracking. This may be another argument in favor of the topological model. However, steric effects behind the RNAP would require that the DNA-binding domain of H–NS rebind the transcribed DNA immediately upstream from RNAP so that it could inhibit the 2-5 nt backtracking movement characteristic of backtracking. We think such immediate, tight rebinding of the transcribed DNA is unlikely. It is possible that Figure 9 could be misinterpreted to suggest this happens, so we have revised the figure to depict H–NS rebinding as leaving space for backtracking.

*4b) The ability of H–NS to increase backtracking needs to be confirmed more thoroughly. Immobilization of the reactions on beads (as in the Rho termination experiment) would allow isolation of paused ECs (as in*
Figure 2*, 16 min lane). Washing away NTPs and then analysis of the susceptibility of paused ECs to Gre (A and/or B) should be carried out. A fast disappearance of pause bands would unambiguously show that the corresponding complexes have backtracked*.

It is unclear to us how this experiment would provide new information relative to that shown in Figure 6 and Figure 6—figure supplement 1. Our data already show that GreB causes H–NS stimulated pause bands to disappear (e.g., the C346 pause in Figure 6). It is unclear to us why immobilizing the complexes, trapping them at the site by washing and NTP deprivation, and then showing that addition of GreB causes the pause bands to disappear provides new information. Indeed, showing the GreB suppression of the pause bands occurs on the time-scale of transcription is a more relevant result than showing that backtracking occurs in halted complexes.

It is possible that the reviewers envision the GreB treatment causing the paused RNAs to get shorter due the RNA cleavage. However, we would have difficulty detecting shortening by 2–5 nt of the RNAs 300-600 nt long using gel electrophoresis. To provide new information of the type sought by the reviewers we have instead established definitively that a key H–NS-stimulated pause, C346, arises by backtracking whereas a key pause not stimulated by H–NS, C370, pauses without evidence of backtracking. We have included these new data as Figure 6—figure supplement 3. To demonstrate backtracking at the H–NS stimulated C346 pause vs lack of backtracking at the non-stimulated C370 pause, we examined pausing at the sequences using much shorter RNAs that could form ECs on DNA scaffolds of appropriate sequence (see Figure 6—figure supplement 3). Strikingly, in the absence of H–NS, no pausing is seen at C346 whereas pausing is seen as C370, especially at 20°C (panel A). Although it would be ideal to test effects of H–NS on these scaffolds, H–NS does not form filaments on these short scaffold DNAs even when high-affinity H–NS sites are added (we have tested many variants of such scaffolds without success, but haven’t included these results in the manuscript in the interest of brevity). Instead, we tested proclivity for backtracking by halting ECs at the C346 or C370 pause sites by NTP deprivation, and then tested for pausing by addition of NTPs, for intrinsic transcript cleavage by adjusting pH to 9 and Mg to 20 mM, and for GreB-stimulated cleavage by addition of GreB (as requested by the reviewers). The results were clear and striking. Halting ECs at the pause dramatically increased the C346 pause but had much less effect on the C370 pause (more EC entered the pause state, but the pause duration was unchanged). This dramatic increase in the C346 pause was accompanied by a significant potential for intrinsic cleavage of the C346 RNA (actually C21 on the scaffold) but not the C370 RNA, and increased potential for GreB-stimulated cleavage of C346 but not C370. By using short RNAs on scaffolds, we could readily detect the cleavage products, and it was clear the C346 pause backtracked by 4–6 nt in the absence of GreB and 4 nt in the presence of GreB (corresponding to 3’ cleavage products of 5–7 nt).

We are happy the reviewers suggested exploring evidence for backtracking further and believe these new results strengthen the manuscript. Clearly a pause with potential to backtrack creates an H–NS stimulated pause, whereas a pause with less potential to backtrack is much less affected by H–NS. To make room for the new results without making the manuscript overlong, we have removed the inconclusive topoisomerase experiment (Figure 5–figure supplement 1) in the original submission.

*5)*
Figure 5
*needs an artist's impression of how the EM data shows bridging on either side of EC*.

We agree and have added an artist’s impression of bridging on either side of an EC to Figure 5.